# Bacterial diversity in arboreal ant nesting spaces is linked to colony developmental stage

Maximilian Nepel [1,2,3 ✉], Veronika E. Mayer [1 ✉], Veronica Barrajon-Santos[1,2,3] & Dagmar Woebken [2]

The omnipresence of ants is commonly attributed to their eusocial organization and division of labor, however, bacteria in their nests may facilitate their success. Like many other arboreal ants living in plant-provided cavities, *Azteca* ants form dark-colored "patches" in their nesting space inside *Cecropia* host plants. These patches are inhabited by bacteria, fungi and nematodes and appear to be essential for ant colony development. Yet, detailed knowledge of the microbial community composition and its consistency throughout the life cycle of ant colonies was lacking. Amplicon sequencing of the microbial 16S rRNA genes in patches from established ant colonies reveals a highly diverse, ant species-specific bacterial community and little variation within an individual ant colony, with Burkholderiales, Rhizobiales and Chitinophagales being most abundant. In contrast, bacterial communities of early ant colony stages show low alpha diversity and no ant species-specific community composition. We suggest a substrate-caused bottleneck after vertical transmission of the bacterial patch community from mother to daughter colonies. The subsequent ecological succession is driven by environmental parameters and influenced by ant behavior. Our study provides key information for future investigations determining the functions of these bacteria, which is essential to understand the ubiquity of such patches among arboreal ants.

[1] Department of Botany and Biodiversity Research, University of Vienna, Vienna, Austria. [2] Department of Microbiology and Ecosystem Science, Centre for Microbiology and Environmental Systems Science, University of Vienna, Vienna, Austria. [3] Doctoral School in Microbiology and Environmental Science, University of Vienna, Vienna, Austria. ✉email: maximilian.nepel@univie.ac.at; veronika.mayer@univie.ac.at

Ants are dominant taxa in terrestrial ecosystems and key ecosystem engineers in tropical rainforests[1,2]. They influence the chemical and physical soil properties by nesting underground and excavating tunnels[3–5], and are important in food chains at higher and lower trophic levels[6,7]. Ants are also involved in plant seed dispersal[8] and, in rare cases, also in plant pollination[9–11].

Commonly known main reasons for the omnipresence of ants are their eusocial organization, with overlapping generations and division of labor, and their well-developed ability to communicate chemically[12]. Yet, as in many animals, associated bacteria may help ants to overcome unfavorable conditions such as pathogenic pressure[13,14] or nitrogen (N)-limitations[15–21]. These bacteria are often closely associated with their ant host and can be found on the body surface or in the gut of ants. Ant-associated Actinobacteriota have been shown to produce antibiotics and are thought to protect the larvae, or, in case of leaf-cutter ants, fungus cultivars against pathogens[13,14]. Furthermore, proteobacterial gut symbionts appear to be particularly associated with herbivorous ants[15,16,21]. They have been shown to recycle N and synthesize essential amino acids[20–22], or have been hypothesized to fix atmospheric dinitrogen ($N_2$)[15–18]. Thus, bacterial mutualists are thought to enable ants to colonize N-limited tropical canopies[7].

Many of the canopy-dwelling ants live in a mutualistic association with plants. The ants defend the host plant against insect herbivores and fungal pathogens[23] and often also provide nutrients to the host plant in form of their debris[24–27]. In exchange, ants receive plant-provided food supply and nesting space such as hollow stems, branches, or leaf pouches (domatia). The few existing studies on bacterial communities in ants inhabiting plants suggest that not only ant body-associated bacteria, but also free-living bacterial communities may play an essential role for the ant colony. Actinobacteriota (Streptomycetales) isolated from domatia of the *Pseudomyrmex penetrator/Tachigali* sp. and *Petalomyrmex phylax/Leonardoxa africana* ant-plant mutualisms, and Proteobacteria (Pseudomonadales) from domatia of the *Azteca* sp./*Cecropia* sp. ant-plant mutualism showed antifungal properties[14,28]. Some strains of *Streptomyces* identified in the domatia of the *P. phylax/L. africana* and *Crematogaster margaritae/Keetia hispida* mutualisms are thought to have the ability to degrade cellulose or to fix $N_2$[14]. Strains of various proteobacterial orders (Rhizobiales, Sphingomonadales, Burkholderiales, Enterobacterales, Pseudomonadales and Xanthomonadales) isolated from domatia of *Azteca* sp. inhabiting *Cecropia* trees and *Allomerus* sp. inhabiting *Hirtella physophora* were suggested to be involved in substrate preparations for the fungus-cultivation activity of these ants[29], in defending against pathogens[28,29] or in $N_2$ fixation[28–30].

Apparently, the bacterial microbiome inside domatia of host trees is considerably influenced by the resident ants. The domatia walls of *Azteca* and *Cephalotes* ants inhabiting the same *Cordia* tree species as a host, contained different bacterial communities, depending on the ant species involved[31]. The walls of domatia inhabited by *Cephalotes* consisted mainly of Actinobacteriota and Proteobacteria, in case of *Azteca* nests also Bacteroidota were abundant[31]. Moreover, *Azteca* domatia walls inside the *Cecropia* host trees harbored unique bacterial communities which differed considerably from those outside the domatia such as the nest entrances or the stem surface of the host plant[32]. Furthermore, the microbiome varied depending on the purpose of each nest chamber[32]. These studies support the notion that ants inhabiting host plants are shaping the microbiome of their domatia[30–32].

Yet, there are still many gaps in our knowledge of the bacterial community inside plants inhabited by mutualistic ants. The studies cited above do not provide details on the consistency of the microbiome of the colonies over time. It is not known whether the microbiome in the domatia of colony founding queens raising her first worker offspring differs from that of the "established ant colonies" that had successfully survived the critical colony-founding phase and had grown to large colonies with numerous workers and often reproduction of new queens and males. It is not known whether microbiome changes occur once the number of workers increases, and more ants forage outside the domatia leading to an increased input of bacteria from the environment. It is not known whether the original bacterial community is retained due to the high concentration of bactericidal and fungicidal chemicals the ants use to keep their nest clean from pathogens[12,33–35]. It is also not yet known whether the microbial community within domatia is specific at the ant genus or even at the ant species level.

To investigate the dynamics of the bacterial community during the life cycle of arboreal ant colonies, we chose one of the most intensively studied ant-plant mutualisms as our model system, the Neotropical *Azteca-Cecropia* association. These *Azteca* ants which reside in the hollow stems (domatia) of *Cecropia* trees, belong to arboreal ant species that consistently form numerous well-defined, dark-colored "patches" in their nesting space, harboring specific Chaetothyriales fungi (Ascomycota)[36–39], bacterivorous nematodes[40,41] and also bacteria[21,22]. *Azteca* foundress queens have been shown to initially form the first patches by scratching off plant parenchyma and inoculating it with patch material from the mother colony, resulting in a vertical transmission from the mother colonies[42]. Subsequently, workers form and maintain new patches throughout the plant. It has been shown that larvae from *Azteca* and various other arboreal ant species take up fungi and other patch material[38,42], indicating a significance for the ant colony. We performed amplicon sequencing of the 16S rRNA genes in patches of three ant colony developmental stages of up to three different *Azteca* ant species to address the following questions: (I) are the bacterial/archaeal community compositions in patches of each ant colony developmental stage *Azteca* species-specific?; (II) is the composition of the bacterial/archaeal community consistent within the patches of the same established *Azteca* colony?; and (III) do the bacterial/archaeal community compositions in patches from early to established ant colony developmental stages increase in diversity, as workers may introduce microorganisms from the plant surface into the nesting space of established colonies, in addition to the otherwise expected vertically transmitted bacterial/archaeal community? Such detailed studies, especially of bacterial/archaeal community dynamics inside the nesting space of arboreal ants, are an important step to further understand their role in ant colony developments.

## Results

Three stages of ant colony development were defined as follows: initial ant colony with the egg laying queen, young colony in a still sealed internode with few workers only, and established colonies inhabiting many adjacent internodes and patrolling the plant surface (Fig. 1). Similarly, the patches were called "initial patch" (IP), "young colony patch" (YP) and "established colony patch" (EP). In general, bacteria dominated the prokaryotic patch communities of all ant colony developmental stages, accounting for 99.9% of total reads. Without removing archaeal taxa from our datasets, we further refer to the bacterial/archaeal patch microbiome as bacterial patch community, as all found patterns and correlations relate to the detected bacterial community.

**Heterogeneous bacterial community composition in patches of early ant colonization stages independent of ant species.** To study if the vertically transmitted bacterial community is ant species-specific and alters as ant colony grows, we investigated

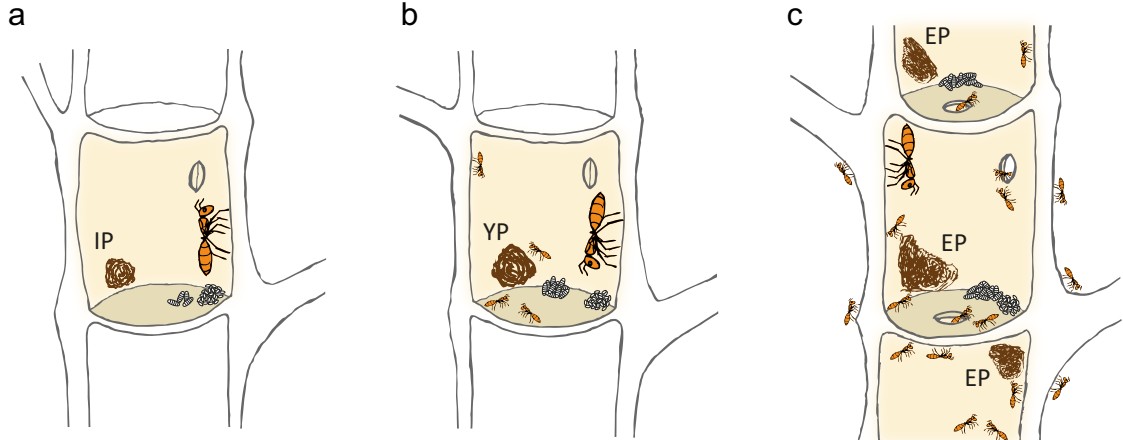

**Fig. 1 Schematics of the three investigated ant colony developmental stages and their corresponding microbial patches. a** The initial ant colony stage consists of the queen, which entered a single plant internode and resealed the entrance hole, brood and an initial pile of scratched-off and chewed plant parenchyma ("initial ant colony patch", IP). **b** In the young ant colony stage, first hatched workers are present in the sealed internode in addition to the queen, brood and patch ("young ant colony patch", YP). **c** In established colonies, the ants inhabit many adjoined plant internodes, patrol the tree surface and typically form numerous patches ("established ant colony patch", EP) throughout their nesting space.

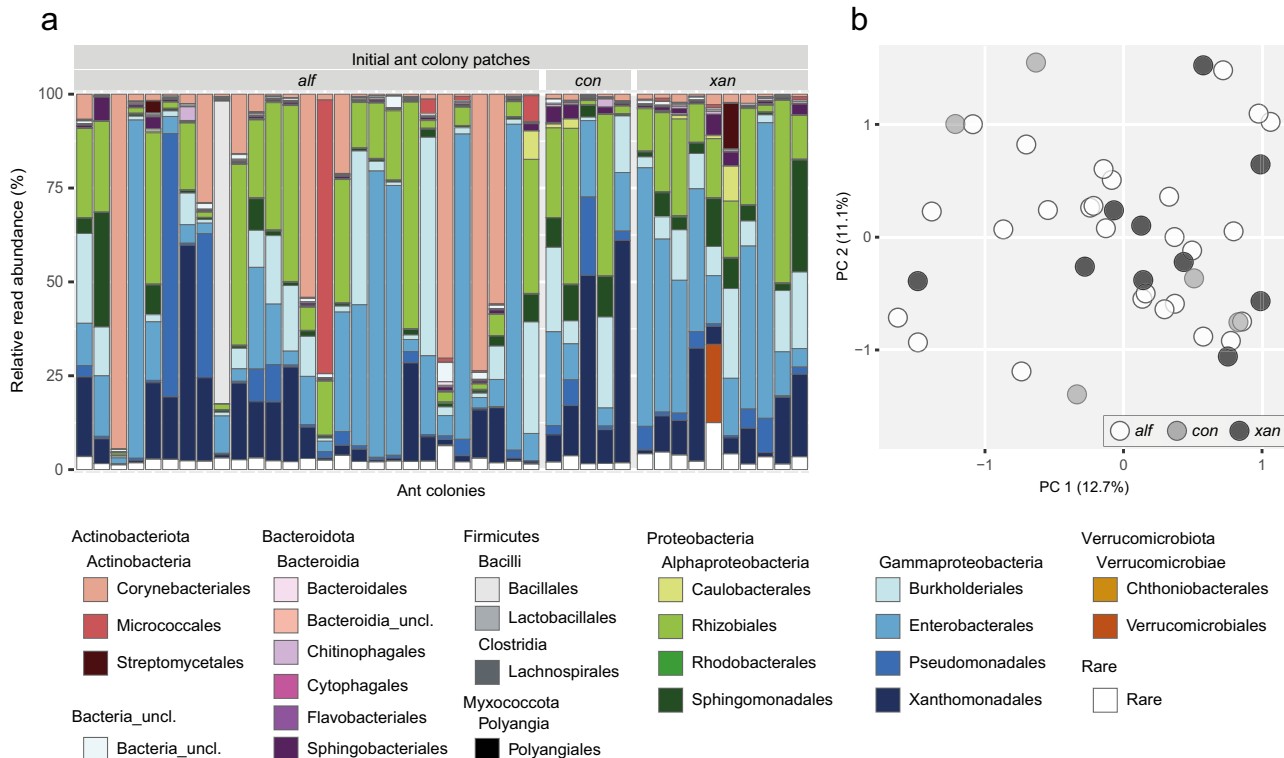

**Fig. 2 Bacterial community composition and beta diversity in patches of initial ant colonies (IPs). a** Taxonomic bar plot depicting the relative read abundance of orders on the y-axis. On the x-axis, every bar represents the bacterial community composition of one IP. **b** Principal coordinates analysis plot, displaying bacterial community similarities amongst IPs. The patch samples are grouped according to the *Azteca* ant species (*alf*: *A. alfari*, n = 27; *con*: *A. constructor*, n = 5; *xan*: *A. xanthochroa*, n = 10).

first the bacterial patch community composition of two early ant colony developmental stages of *Azteca alfari*, *A. constructor* and *A. xanthochroa*. These closely related ant species were found to colonize *Cecropia* plants at our study site in Costa Rica. We analyzed patches of 42 initial ant colonies (IPs) surrounded by the founding queen, eggs, brood (Fig. 1a), and from patches of 18 young ant colonies (YPs) surrounded by already hatched workers additionally to the queen, eggs and brood (Fig. 1b). In both cases, each colony was restricted to a single, sealed plant internode.

In the bacterial communities of IPs, Gammaproteobacteria (on average 54.9%), Alphaproteobacteria (23.9%) and Actinobacteria (14.0%) showed highest mean relative read abundances (Fig. 2a). A strong heterogeneity in bacterial community composition across IP samples could be seen on taxonomic order level with Enterobacterales (on average 25.9%, ranging from 1.5–90.1%), Rhizobiales (18.5%, 0.6–60.3%) and Xanthomonadales (13.1%, 0.2–59.2%) showing highest mean relative abundance of reads, followed by Corynebacteriales (11.0%, 0.2–94.5%) and

**Table 1 Prevalent ASVs of initial ant colony patches (IPs) and young ant colony patches (YPs), each accounting for more than 0.4% of reads in more than 50% of IP or YP samples, respectively.**

| | Class | Order | Family | ASV | Av.ab.[a] |
|---|---|---|---|---|---|
| IPs | Alphaproteobacteria | Rhizobiales | *Rhizobiaceae* | ASV_5 | 8.39 |
| . | . | . | . | ASV_26 | 4.17 |
| . | Gammaproteobacteria | Enterobacterales | Enterobacterales_unclassified | ASV_4 | 9.13 |
| . | . | . | *Enterobacteriaceae* | ASV_1 | 2.98 |
| . | . | . | *Erwiniaceae* | ASV_14 | 6.87 |
| YPs | Alphaproteobacteria | Rhizobiales | *Rhizobiaceae* | ASV_5 | 13.04 |
| . | . | . | . | ASV_26 | 3.04 |
| . | . | Sphingomonadales | *Sphingomonadaceae* | ASV_30 | 2.11 |
| . | . | . | . | ASV_54 | 2.56 |
| . | Gammaproteobacteria | Burkholderiales | *Oxalobacteraceae* | ASV_15 | 5.71 |
| . | . | Enterobacterales | Enterobacterales_unclassified | ASV_4 | 3.64 |
| . | . | . | *Enterobacteriaceae* | ASV_1 | 1.44 |
| . | . | . | *Erwiniaceae* | ASV_14 | 1.89 |
| . | . | Xanthomonadales | *Xanthomonadaceae* | ASV_8 | 2.89 |

[a]Av.ab.... average relative abundance (%) of all reads from initial or young ant colony patches, respectively.

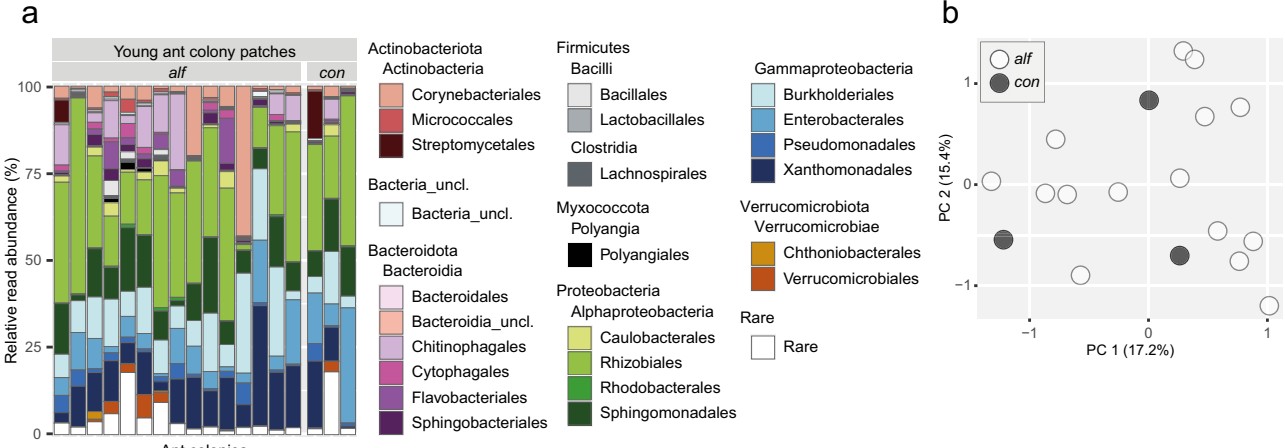

**Fig. 3 Bacterial community composition and beta diversity in patches of young ant colonies (YPs). a** Taxonomic bar plot depicting the relative read abundance of orders on the y-axis. On the x-axis, every bar represents the bacterial community composition of one YP. **b** Principal coordinates analysis plot, displaying bacterial community similarities amongst YPs. The patch samples are grouped according to the *Azteca* ant species (*alf*: *A. alfari*, n = 15; *con*: *A. constructor*, n = 3).

Burkholderiales (10.5%, 0.5–58.2%). The variation in community composition did not correlate with the three ant species (*P* = 0.178) and thus, communities did not cluster according to ant species in the beta diversity ordination plot (Fig. 2b). Few ASVs accounted for a considerable amount of amplicon reads. Five of around 3400 ASVs were defined as prevalent – ASVs that account for more than 0.4% of reads in more than 50% of respective samples. These prevalent ASVs belonging to Proteobacteria were present in every single IP and accounted for 31.5% of the total IP reads. These ASVs were assigned to Rhizobiales, and Enterobacterales (Table 1).

In YPs, Alphaproteobacteria (on average 40.9%) and Gammaproteobacteria (35.8%) accounted for highest relative read abundances, followed by Bacteroidia (9.8%) and Actinobacteria (8.9%) (Fig. 3a). Compared to IPs, the bacterial communities across YP samples were less heterogeneous. The most prominent orders were Rhizobiales (on average 27.9%, ranging from 1.5–56.2%), Xanthomonadales (12.0%, 0.6–34.4%), Burkholderiales (11.6%, 2.5–28.7%) and Sphingomonadales (10.7%, 1.5–21.7%). The variation in beta diversity of YPs did not correlate with the two ant species (*P* = 0.740) (Fig. 3b). Nine

prevalent ASVs accounted for 36.3% of total YP reads and were assigned to Rhizobiales, Sphingomonadales, Burkholderiales, Enterobacterales and Xanthomonadales (Table 1).

**Little variation in bacterial patch community composition within established ant colonies.** Prior to investigating the ant species-specificity of bacterial patch communities across established ant colonies, the variation of bacterial communities within an established ant colony had to be assessed. Within established *A. alfari* and *A. constructor* colonies, ant-built patches (EPs) are typically found throughout the nesting space (Fig. 1c). We observed different ages of such patches during field work, which was linked to their spatial location in the *Cecropia* tree. Patches from the youngest growth zone of the tree (the apical stem internodes) (EP I) were only recently formed. Patches from the middle (EP II) and lower (EP III) part of the ant colonies were gradually older. To investigate the variation of the bacterial community compositions along this age gradient, we repeatedly sampled EPs throughout the nesting space of 17 *Azteca* ant colonies and sequenced each EP separately. The alpha diversity in EPs was similar and did not significantly differ from younger to

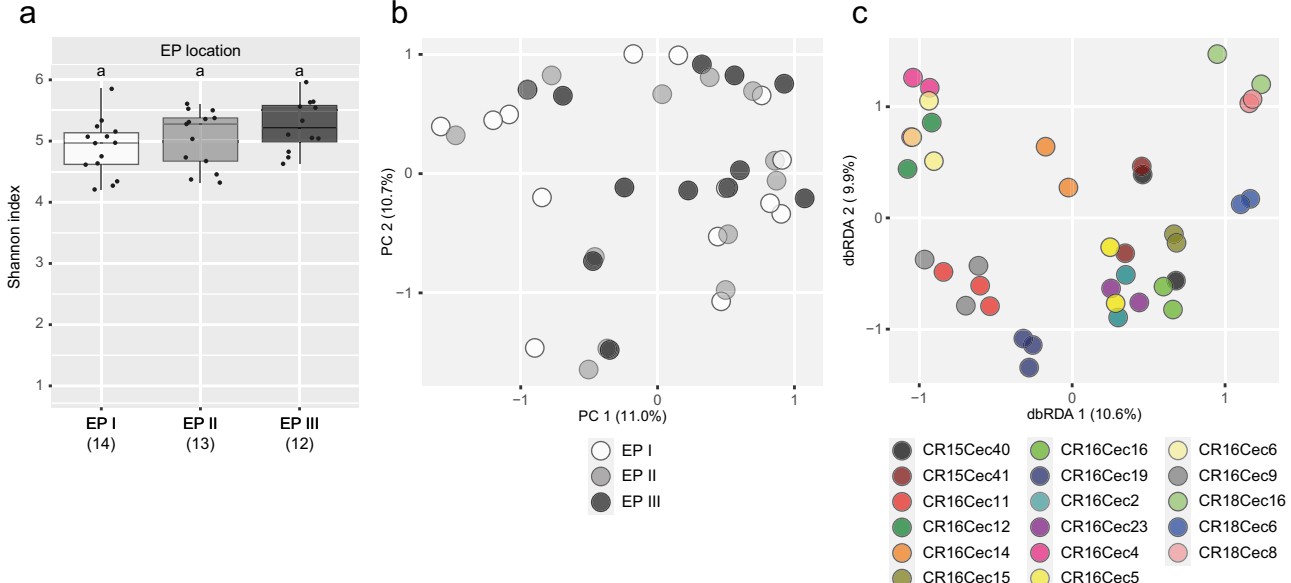

**Fig. 4 Alpha and beta diversity of bacterial communities in patches differently located in *Cecropia* stems of established ant colonies.** Recently formed patches from apical stem internodes (EP I, $n = 14$), patches from the middle (EP II, $n = 13$) and lower (EP III, $n = 12$) actively colonized stem parts were distinguished, representing an age gradient of patches. **a** Shannon indices, depicting the alpha diversity in EPs within established ant colonies. Sampling sizes are given in brackets. Letters denote no significant differences between patch types. **b** Principal coordinates analysis plot, displaying bacterial community dissimilarities between EPs of presumed age. **c** Distance-based redundancy analysis plot, showing variation of bacterial communities in EPs due to the ant colonies they originated from. Please note that in this analysis, data are not differentiated between ant species.

older patches (Fig. 4a, $P = 0.125$; median Shannon index: EP I 4.9, EP II 5.3, EP III 5.2). Also, the variation in community composition did not correlate with the age of the patches ($P = 0.995$; $P = 0.923$, if permutations were set within the ant species) and therefore samples of the presumably same age group did not cluster together in beta diversity ordination plots (Fig. 4b). EPs of the same ant colony, but of different age, had a similar community composition (Fig. 4c), which was reflected by the significant correlation between the beta diversity and the ant colony ID ($P < 0.001$), explaining 73.0% of the variation.

**Ant species-specific bacterial patch communities in established ant colonies.** By studying our extended dataset of 34 established ant colonies, we uncovered a high bacterial diversity. We combined the samples of the 17 previously mentioned established ant colonies that were repeatedly sampled with 17 ant colonies of which only one patch sample was taken. Generally, ASVs were assigned to more than 25 phyla and more than 80 classes. Gammaproteobacteria (on average 28.6%), Bacteroidia (23.7%) and Alphaproteobacteria (23.5%) showed the highest relative read abundances in patches of established ant colonies, followed by Verrucomicrobiae (5.2%) and Actinobacteria (5.2%). The most read-abundant orders were Burkholderiales (on average 17.6%), Chitinophagales (12.5%) and Rhizobiales (10.9%) (Fig. 5a). Significant differences in the bacterial community between the ant species could be detected. EPs of *A. constructor* were significantly more diverse than those of *A. alfari* ($P = 0.003$; median Shannon indices: 4.8 *A. alfari*, 5.4 *A. constructor*; Fig. 5b). Also, the beta diversity correlated significantly with the ant species, explaining 9.0% of the variation ($P < 0.001$, Fig. 5c). While the mean relative abundance of several taxonomic orders differed visually between both ant species (Fig. 5a, e.g. Chitinophagales, Rhizobiales, Burkholderiales), some changes were statistically significant. Mainly Rhizobiales (*A. alfari*: 16.1%, *A. constructor*: 8.0%), Verrucomicrobiales (7.7% vs. 2.2%), Xanthomonadales (6.6% vs. 4.0%) and Corynebacteriales (1.9% vs. 1.1%) displayed a significantly higher relative read abundance in EPs of *A. alfari* than of *A.*

*constructor* (Supplementary Table 1). Fourteen of around 8400 ASVs were prevalent in either *A. alfari* or *A. constructor* EPs and made up 18.6% of total EP reads. Despite significant differences in the community composition between ant species, three ASVs were prevalent in both ant species. These ASVs were assigned to Burkholderiales and Sphingomonadales (Table 2). ASVs that reached the prevalence threshold in patches of only one of the two ant species belonged to a variety of different orders: Corynebacteriales, Rhizobiales, Burkholderiales, Xanthomonadales, Verrucomicrobiales, Micrococcales, Chitinophagales and Rhodobacterales.

**Bacterial community composition changed along developmental stages of ant colonies.** By combining all bacterial patch communities of initial, young and established ant colonies – one community per colony – sequencing data covered three stages of ant colony development. This allowed us to analyze changes in the bacterial community composition along the life cycle of *Azteca* colonies. The alpha diversity of patches (regardless of ant species) correlated with the ant colony developmental stages ($P < 0.001$, $x^2 = 70.483$) and increased significantly from IPs via YPs to EPs (pairwise, all $P < 0.001$; median Shannon indices: IPs 2.2, YPs 3.5, EPs 5.2; Fig. 6a). Taking the ant species into account, the same pattern was generally visible except for *A. constructor*, in which IPs and YPs did not significantly differ (Supplementary Fig. 1). Furthermore, the variation in beta diversity was significantly correlating with the ant colony developmental stages explaining 15.4% of the variation ($P < 0.001$; Fig. 6b). Certain taxonomic classes and orders increased and decreased in the average relative read abundance from IPs via YPs to EPs. Bacteroidia (IP: 2.0%, YP: 9.8%, EP: 22.2%), Clostridia (0.3%, 0.3%, 3.1%) and Verrucomicrobiae (0.7%, 1.4%, 6.0%) increased, while the read relative abundance of Actinobacteria (14.0%, 8.9%, 5.3%) and Gammaproteobacteria (54.9%, 35.8%, 29.4%) decreased (Supplementary Fig. 2). Taxonomic orders tended to increase (e.g. Chitinophagales, Flavobacteriales, Verrucomicrobiales) or decrease

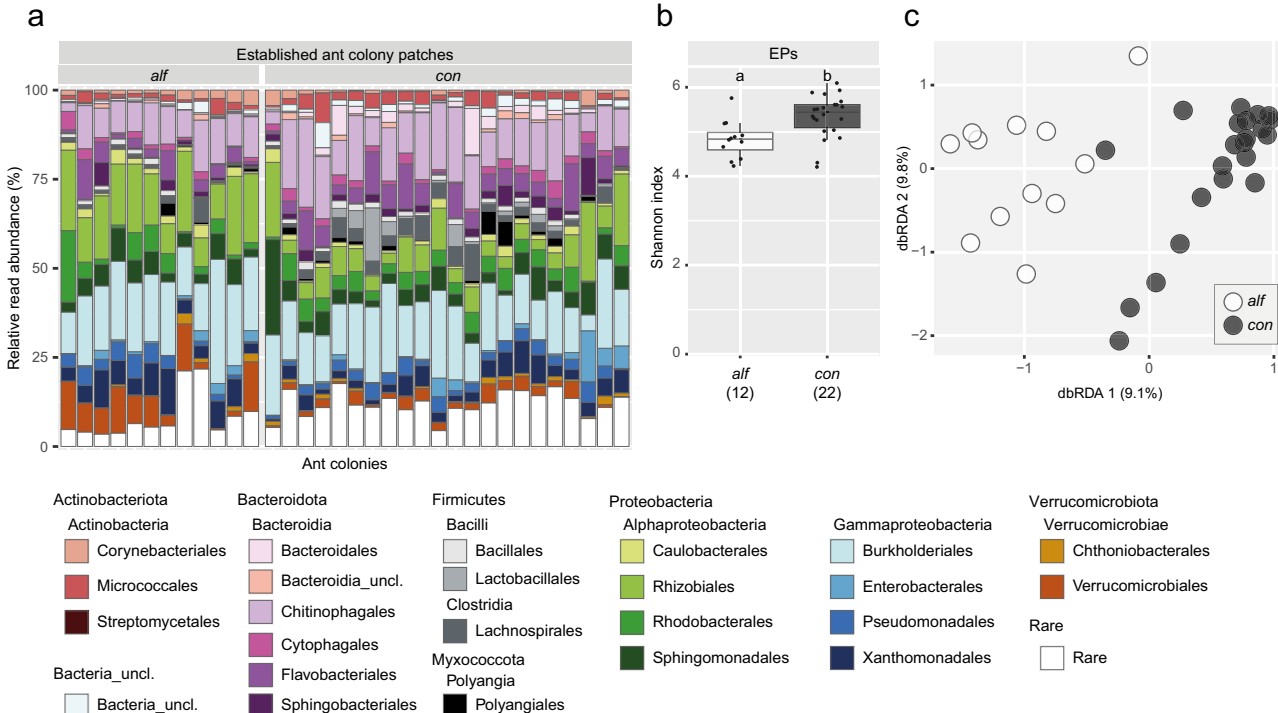

**Fig. 5 Bacterial community composition, alpha and beta diversity in patches of established ant colonies (EPs). a** Taxonomic bar plot depicting the relative read abundance of orders on the y-axis. On the x-axis, every bar represents the bacterial community composition of one ant colony. **b** Shannon indices, depicting the alpha diversity in EPs per ant species. Sampling sizes are given in brackets. Letters denote significant differences between sample types. **c** The distance-based redundancy analysis plot shows the variation of bacterial communities in EPs due to ant species. The patch samples are grouped according to the *Azteca* ant species (*alf*: *A. alfari*, n = 12; *con*: *A. constructor*, n = 22). Please note that this dataset comprises average bacterial communities of established ant colonies that were spatially sampled multiple times (Fig. 4), and bacterial communities of established ant colonies that were only sampled once.

---

**Table 2 Prevalent ASVs of established ant colony patches (EPs), each accounting for more than 0.4% of total reads in more than 50% of either *A. alfari* or *A. constructor* EPs.**

|  | Class | Order | Family | ASV | Av.ab.[a] |
|---|---|---|---|---|---|
| Both | Alphaproteobacteria | Sphingomonadales | *Sphingomonadaceae* | ASV_30 | 0.97 |
| . | Gammaproteobacteria | Burkholderiales | *Comamonadaceae* | ASV_12 | 2.01 |
| . | . | . | . | ASV_6 | 3.44 |
| *A. alfari* | Actinobacteria | Corynebacteriales | *Nocardiaceae* | ASV_11 | 0.26 |
| . | Alphaproteobacteria | Rhizobiales | *Rhizobiaceae* | ASV_5 | 1.84 |
| . | . | . | . | ASV_9 | 1.07 |
| . | . | . | . | ASV_35 | 1.12 |
| . | Gammaproteobacteria | Burkholderiales | *Comamonadaceae* | ASV_21 | 0.86 |
| . | . | . | . | ASV_41 | 0.66 |
| . | . | Xanthomonadales | *Xanthomonadaceae* | ASV_8 | 0.71 |
| . | Verrucomicrobiae | Verrucomicrobiales | *Rubritaleaceae* | ASV_32 | 1.42 |
| *A. constructor* | Actinobacteria | Micrococcales | Micrococcales_unclassified | ASV_37 | 0.62 |
| . | Bacteroidia | Chitinophagales | *Chitinophagaceae* | ASV_10 | 2.16 |
| . | Alphaproteobacteria | Rhodobacterales | *Rhodobacteraceae* | ASV_18 | 1.44 |

[a]Av.ab.… average relative abundance (%) of all reads from established ant colony patches.
The first column denotes if the ASV fulfilled the criteria in patches of one or both ant species. Please note that one ASV listed as prevalent in EPs of one ant species does not necessarily indicate its absence in the other ant species, only being less abundant than our defined threshold. Please note that this dataset comprises average bacterial communities of established ant colonies that were spatially sampled multiple times (Fig. 4), and bacterial communities of established ant colonies that were only sampled once.

---

(e.g. Enterobacterales, Xanthomonadales, Corynebacteriales) in their relative read abundance from early to late developmental stages (Fig. 6c), of which several shifts from IPs to YPs, or many from YPs to EPs were statistically significant (Table 3).

In total, 20 ASVs were identified as prevalent in either IPs, YPs, or EPs (either *A. alfari*, or *A. constructor*) making up 39.8% of total reads. One ASV (assigned to Rhizobiales) reached the prevalence threshold in all three developmental stages (Table 4,

Supplementary Fig. 3). Four ASVs assigned to Enterobacterales and Rhizobiales were prevalent in both IPs and YPs. Two ASVs assigned to Sphingomonadales and Xanthomonadales were prevalent in both YPs and EPs. Two and eleven ASVs reached the prevalence threshold only in either YPs or EPs, whereas no ASV was exclusively prevalent in IPs, or exclusively shared between IPs and EPs. The distribution and abundance of these prevalent ASVs showed particular patterns. Many of these

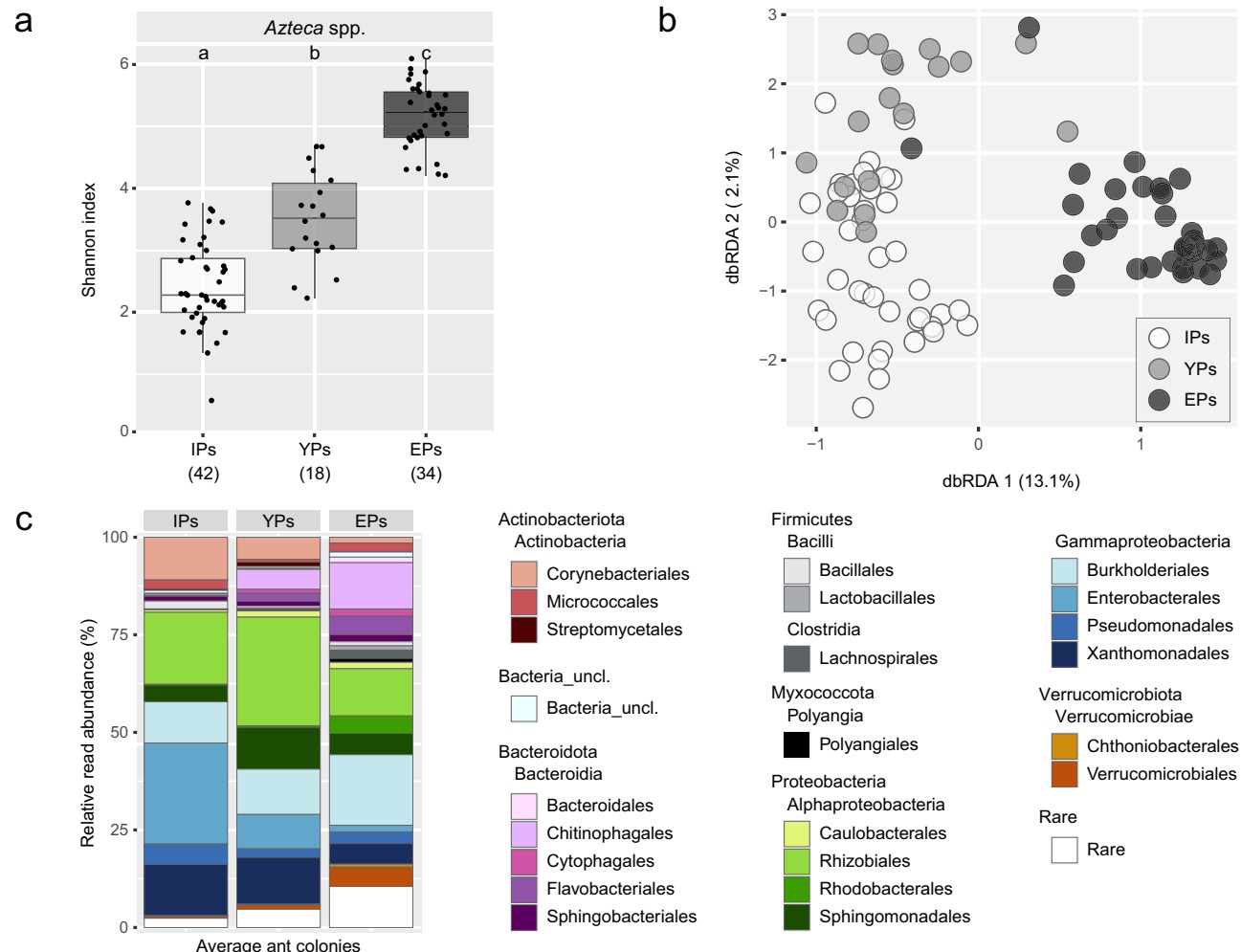

**Fig. 6 Changing bacterial community composition, alpha and beta diversity along developmental stages of ant colonies, including patches of initial (IP), young (YP) and established (EP) ant colonies. a** Shannon indices, depicting the alpha diversity in patches of all ant colony developmental stages independent of the *Azteca* ant species. Sampling sizes are given in brackets. Letters denote significant differences between patch types. **b** Distance-based redundancy analysis plot, showing the variation of bacterial communities in patches due to ant colony developmental stages. **c** Taxonomic bar plot depicting the relative read abundance of orders on the y-axis. On the x-axis, every bar represents the average bacterial community composition per ant colony developmental stage. Please note that for established ant colonies that were spatially sampled multiple times (Fig. 4) an average bacterial community was calculated. Please also note that displaying average community compositions per ant colony developmental stage (**c**) is a simplification due to the heterogeneity across patch samples (see Figs. 2a, 3a and 5a).

prevalent ASVs were most abundant in IPs and YPs, or almost exclusively detected in EPs (Supplementary Fig. 4). Few ASVs were rather equally abundant in both YPs and EPs, but there was a lack of prevalent ASVs being abundant in both IPs and EPs, but not in YPs.

## Discussion
Ant-built patches are commonly found in the nesting space of various ant-plant associations[36,37,39]. They are assumed to play important roles for ant colonies, such as housing bacteria with capabilities against pathogens[14,28], or providing nutrients to ant larvae[38,42]. In addition, the patches potentially enrich the system with N via atmospheric $N_2$-fixing bacteria[30]. Yet, a detailed analysis of the bacterial community, its consistency during the life cycle of the ant colony and its ant species-specificity was lacking so far.

### Bacterial succession along developmental stages of ant colonies. Our study shows that ant species-specificity of the bacterial communities in the patches is determined by the developmental

stage of the *Azteca* colony (Fig. 1). While the bacterial community composition is quite homogeneous across patches of established *A. alfari* or *A. constructor* colonies (EPs) and ant species-specific (Fig. 5), at early stages of ant colony development (IPs or YPs), the bacterial community was highly heterogeneous and showed no correlation with the corresponding ant species (Figs. 2 and 3). Although it has been shown that alate founding queens vertically transmit patch organisms when they colonize a new host plant and form the initial patch[42], the bacterial community composition of the mother colony does not appear to be maintained. The alpha diversity increased from low diverse IPs, via YPs to highly diverse EPs (Fig. 6a), which was only expected in established ant colonies as workers may introduce surface bacteria[43]. The community composition in YPs appeared to be an intermediate stage between IPs and EPs (Fig. 6b, c), with several shared ASVs being prevalent in either IPs and YPs, or YPs and EPs, but no ASV being prevalent in both, IPs and EPs (Supplementary Fig. 4). The factors influencing the bacterial community during the ant colony development may be manifold, but the most important impact may be the N-depleted and cellulose-dominated initial patch

**Table 3 Mean relative read abundance (%) of selected taxonomic orders per ant colony developmental stages.**

| Selected orders | Class | IP | | YP | | *alf* | *con* | EP |
|---|---|---|---|---|---|---|---|---|
| Corynebacteriales | Actinobacteria | 11.0 | | 5.7 | ↘ | ** | ** | 1.5 |
| Micrococcales | Actinobacteria | 2.3 | | 0.7 | | | | 2.2 |
| Streptomycetales | Actinobacteria | 0.4 | | 1.2 | ↘ | | *** | 0.1 |
| Chitinophagales | Bacteroidia | 0.4 | | 5.1 | | | | 11.9 |
| Cytophagales | Bacteroidia | 0.09 | ↗* | 1.0 | | | | 1.8 |
| Flavobacteriales | Bacteroidia | 0.09 | ↗*** | 2.1 | | | | 4.8 |
| Sphingobacteriales | Bacteroidia | 1.2 | | 1.3 | ↗ | * | ** | 1.8 |
| Caulobacterales | Alphaproteobacteria | 0.6 | | 1.6 | | | | 1.6 |
| Rhizobiales | Alphaproteobacteria | 18.5 | | 27.9 | ↘ | ** | *** | 12.0 |
| Rhodobacterales | Alphaproteobacteria | 0.2 | | 0.4 | ↗ | ** | * | 4.5 |
| Sphingomonadales | Alphaproteobacteria | 4.3 | ↗*** | 10.7 | ↘ | ** | *** | 5.4 |
| Burkholderiales | Gammaproteobacteria | 10.5 | | 11.6 | ↗ | | *** | 18.1 |
| Enterobacterales | Gammaproteobacteria | 25.9 | ↘* | 8.8 | ↘ | *** | *** | 1.7 |
| Pseudomonadales | Gammaproteobacteria | 5.2 | | 2.2 | ↗ | | ** | 3.0 |
| Xanthomonadales | Gammaproteobacteria | 13.1 | | 12.0 | ↘ | ** | *** | 5.3 |
| Verrucomicrobiales | Verrucomicrobiae | 0.5 | | 1.1 | ↗ | ** | | 4.9 |

Patches of initial (IP), young (YP) and established (EP) ant colonies were distinguished. Asterisks denote statistically significant changes (↗ increase, ↘ decrease) in relative read abundances between IPs-YPs and YPs-EPs (*** < 0.001, ** < 0.01, * < 0.05). Due to ant species-specific community compositions in EPs, YPs were compared separately with EPs of *A. alfari* (*alf*), or *A. constructor* (*con*). Please note that for established ant colonies that were spatially sampled multiple times (Fig. 4) an average bacterial community was calculated. Please also note that the average relative abundance per ant colony development stage is a simplified point of view. The observed heterogeneity across patch samples can be extracted from Figs. 2a, 3a and 5a.

**Table 4 Occurrence of prevalent ASVs per ant colony developmental stages.**

| | Class | Order | Family | ASV | Av.ab.[a] |
|---|---|---|---|---|---|
| All_3 | Alphaproteobacteria | Rhizobiales | *Rhizobiaceae* | ASV_5 | 6.91 |
| IP_YP | Alphaproteobacteria | Rhizobiales | *Rhizobiaceae* | ASV_26 | 2.47 |
| . | Gammaproteobacteria | Enterobacterales | Enterobacterales_unclassified | ASV_4 | 4.93 |
| . | . | . | *Enterobacteriaceae* | ASV_1 | 1.83 |
| . | . | . | *Erwiniaceae* | ASV_14 | 3.45 |
| YP_EP | Alphaproteobacteria | Sphingomonadales | *Sphingomonadaceae* | ASV_30 | 0.97 |
| . | Gammaproteobacteria | Xanthomonadales | *Xanthomonadaceae* | ASV_8 | 3.20 |
| YP | Alphaproteobacteria | Sphingomonadales | *Sphingomonadaceae* | ASV_54 | 0.91 |
| . | Gammaproteobacteria | Burkholderiales | *Oxalobacteraceae* | ASV_15 | 3.40 |
| EP | Actinobacteria | Corynebacteriales | *Nocardiaceae* | ASV_11 | 2.42 |
| . | . | Micrococcales | Micrococcales_unclassified | ASV_37 | 0.23 |
| . | Bacteroidia | Chitinophagales | *Chitinophagaceae* | ASV_10 | 0.79 |
| . | Alphaproteobacteria | Rhizobiales | *Rhizobiaceae* | ASV_9 | 1.04 |
| . | . | . | . | ASV_35 | 0.44 |
| . | . | Rhodobacterales | *Rhodobacteraceae* | ASV_18 | 0.53 |
| . | Gammaproteobacteria | Burkholderiales | *Comamonadaceae* | ASV_21 | 0.53 |
| . | . | . | . | ASV_41 | 0.25 |
| . | . | . | . | ASV_12 | 0.98 |
| . | . | . | . | ASV_6 | 1.28 |
| . | Verrucomicrobiae | Verrucomicrobiales | *Rubritaleaceae* | ASV_32 | 0.52 |

[a]Av.ab.... average relative abundance (%) of total reads.
Each ASV is accounting for at least 0.4% of reads in more than 50% of either initial (IP), young (YP) or established (EP, either *A. alfari*, or *A. constructor*) ant colonies. The first column displays in which ant developmental stage the ASV fulfilled the criteria. Please note that one ASV listed as prevalent in one patch type does not necessarily indicate its absence in the other types, only being less abundant than our defined threshold. Please also note that for established ant colonies that were spatially sampled multiple times (Fig. 4) an average bacterial community was calculated.

substrate. This obviously causes a bottleneck during the colony foundation and induces an ecological succession after vertical transmission of the highly diverse bacterial EP community.

The drivers of the succession are primarily biotic and abiotic patch conditions. Personal field observations indicate changing substrate for bacteria when IPs develop into YPs and over time into EPs. While the substrate of early patches (IPs, YPs) exclusively consists of N-poor *Cecropia* parenchyma (average C/N ratio of around 180, Supplementary Table 2), the substrate in established patches is more diverse. When workers open the sealed domatia entrance and start to forage on the surface of the host plant, they also place items like plant-provided food bodies, parts of mosses and lichens on EPs. While the colony grows, workers die and the fragmented exoskeletons of the dead nest

mates are placed on the patches as well. With the latter, the ants of established colonies enrich the patches with a high amount of chitin as a potential N-source which considerably changes the substrate. Additionally, we even visually observed an increase of fungal and nematode biomass from IPs to YPs and to EPs. Fungi and nematodes probably not only introduce their own associated bacteria to the patch community, but also a variety of usable compounds, either by secretion (fungi), feces deposition (nematodes), or decay of their biomass. An important, but often neglected aspect is the behavior of the ants to reduce the pathogen pressure in the nest, which may also impact the bacterial community. This can be done either by fumigating the nesting space with antibiotic compounds from, for example, the metapleural gland[44] or by lowering the pH, as shown for leaf

cutter ants[45,46]. The change from a crumbly patch texture in IPs to a denser texture in EPs is also likely to change oxygen and water content[30]. This certainly affects some members of the bacterial community. In addition to the changing conditions in patches, various other sources of bacteria may alter the bacterial diversity along the ant colony life cycle. While the colony grows, ant workers start foraging on the plant surface and thereby bringing bacteria on the tarsi structure[43] into the nest. Workers also colonize new internodes that may house patches with a different microbiome from unsuccessful colony founding attempts where the queens had already died[42].

To see if the identified bacterial community reflects a patch condition-driven succession, we identified cultivated strains in GenBank (NBCI) with high 16S rRNA sequence similarity to our prevalent bacterial ASVs (Table 4). Selected ASVs prevalent in IPs and/or YPs, which are parenchyma-dominated and therefore N-depleted, showed 100% 16S rRNA sequence similarity with $N_2$-fixing bacteria (ASV_15, Burkholderiales, *Herbaspirillum frisingense*[47]), with cellobiose-utilizing bacteria (ASV_54, Sphingomonadales, *Sphingobium yanoikuyae*[48]), or with bacteria generally associated with plant tissue (ASV_26, Rhizobiales, *Agrobacterium vitis*[49]). In EPs where the turnover of various patch organisms and dead ant bodies – all commonly containing glycogen as a storage compound – might play a role, ASV_10 (Chitinophagales) was prevalent. The cultivated strain with high sequence similarity to this ASV (*Terrimonas aquatica* 97.6% 16S similarity) has been shown to use glycogen and cellobiose as carbon sources[50]. However, further cultivation- or metagenomics-based investigations are needed to determine the metabolic potential and thus probable functions of these bacteria in patches. This might shed light on the question why they are abundant in certain ant colony developmental stages and not in others.

**Intraspecific variation of bacterial community composition amongst - but not within - established *Azteca* colonies.** The substrate-driven succession may not only have caused the transformation of the bacterial patch community from initial to established ant colonies, but also may have led to different bacterial communities within an established ant colony. Newly formed EPs in newly grown shoots at the apical end of the plant are comparable to IPs as they consist primarily of plant parenchyma which is still available in uncolonized "new" internodes. We therefore assumed that the bacterial community in the patches of established colonies would undergo comparable transformation processes within the stem of a host plant as between the founding stage and established colony stages, and that the patch community would vary considerably between younger and older EPs. However, we showed that the bacterial communities of all patches within an established ant colony were similar to each other, even though the patches were of different age (Fig. 4). This suggests that workers maintain a specific bacterial community throughout the entire nesting space by inoculating the newly built patches with material from older patches. Field observations with *A. constructor* colonies (by MN and VEM) showed additional evidence of such patch management. There was a lack of patch material in abandoned internodes. In the oldest, still actively colonized internodes, there were globules of ant-made patch material, probably used to inoculate new patches in the apical part of the nesting space.

**Diversity of bacterial patch communities in established *Azteca* colonies are driven by ant species.** We have convincingly demonstrated that the respective ant species plays a major role in shaping the bacterial communities in established *Azteca* colonies.

Previous studies have shown differences in the microbiome between ant colonies of different genera[31], and between ants' nests and their surroundings[32]. Our study takes a step forward and shows that the bacterial communities associated with the nests differ even among two closely related ant species: *A. constructor* maintained a significantly different and more diverse bacterial community than *A. alfari* (Fig. 5). This indicates that the ants influence the bacterial community of a patch through ant species-specific behavior. In established ant colonies, *Azteca* species form significantly different patch morphologies, which may explain the variation within the bacterial community. The EPs of *A. constructor* are thought to contain more micro niches due to their typical pile structure which allows the development of moisture and gas gradients, whereas *A. alfari* EPs are more lawn-like[30]. Interestingly matching the $O_2$ gradient towards anoxic conditions in EPs of *A. constructor* EPs[30], a cultivated strain in GenBank has been described as facultative anaerobic, which has a high sequence similarity to an ASV frequently found in *A. constructor* EPs (Table 2) (ASV_37, Micrococcales, 98.4% 16S rRNA gene similarity to *Changpingibacter yushuensis*[51]). In addition, the differences in alpha diversity support the idea that the pile-shaped EPs of *A. constructor* experience gradients of humidity and chemical concentrations, facilitating a more diverse bacterial community than in *A. alfari* EPs. Though, further research is needed to verify chemical gradients and their influence on the bacterial community composition, or to investigate if ant species-specific chemistry like scent or other excretions may shape the bacterial communities.

In summary, we provide the first evidence of a dynamic bacterial patch community within arboreal ant nests. The bacterial communities in these patches appear to undergo an ecological succession within each ant colony, thus evolving with the ant colony's life cycle. Following an assumed vertical transmission of bacteria from mother to daughter colonies, the composition of the bacterial community seems to experience a bottleneck caused by the substrate. This bottleneck initially favors bacteria adapted to N-depleted cellulose-dominant substrates in early colonization stages. The subsequent ecological succession is driven by ant behavior, substrate changes and environmental parameters, and results in ant species-specific bacterial communities. We conceptualize patches inside ant-plant associations as complex ecosystems, which are essential for the development of the inhabiting ant colony. This assumption is based on the unexpectedly high bacterial diversity within colonized host plants, and the supposed diverse functional potentials in the patches. Our findings provide crucial insights for future cultivation- and omics-based studies, which can help to determine the role that these patches play in ant colonies.

## Material and methods
**Study site characteristics and sample collection.** *Cecropia* plants (Urticaceae) colonized by three *Azteca* species (*A. alfari*, *A. constructor* and *A. xanthochroa*; Formicidae, Dolichoderinae) were investigated between 2015 and 2018 in close vicinity to the Tropical Research Facility La Gamba, Costa Rica (www.lagamba.at; N08°42'03", W083°12'06", 70 m asl). In total, we investigated 116 patches from 65 *Cecropia* plants of different sizes – 31 saplings and 34 trees – located at the margins of primary and secondary rain forests, alongside rivers (Rio Bonito, Rio Sardinal) and roads. The sampling procedure has been described in Nepel et al.[30]. Due to the necessity of destructively cutting *Cecropia* saplings and trees prior to assessing the presence, developmental stage and species of inhabiting ant colonies, sample sizes vary. We differentiated between three ant colony developmental stages and their corresponding patches (Fig. 1). For claustral colony

foundation, the queen had entered a single plant internode, resealed the entrance hole and formed an initial pile of scratched-off and chewed plant parenchyma ("initial ant colony patch", IP, Fig. 1a). In an intermediate stage, the still sealed internode sheltered a young ant colony consisting of queen, brood and the first hatched workers, next to the patch ("young ant colony patch", YP, Fig. 1b). In established colonies, the ants inhabited many adjoined internodes of the stem, patrolled the whole tree and typically formed numerous defined dark-colored patches ("established ant colony patch", EP, Fig. 1c) throughout their nesting space.

We sampled 31 young *Cecropia* saplings that were recently colonized by *Azteca* ants, of which 60 single internodes were just inhabited by foundress queens with IPs or optionally first workers, in which case these patches were defined as YPs. From the plant saplings we sampled in total 42 IPs (*A. alfari* $n = 27$, *A. constructor* $n = 5$, and *A. xanthochroa* $n = 10$), and in total 18 YPs (*A. alfari* $n = 15$ and *A. constructor* $n = 3$). Young and established *A. xanthochroa* colonies were not found in our study area.

We sampled 34 *Cecropia* trees, which were inhabited by established *A. alfari* ($n = 12$) and *A. constructor* ($n = 22$) ant colonies. To test if the bacterial/archaeal community composition remains stable in patches of the same established ant colony, patches from five of the *A. alfari* and twelve of the *A. constructor* colonies were collected separately from up to three stem parts representing different ant colony activity and patch age: EPs of the apical plant area recently colonized by the ants and formed with a lot of parenchyma as substrate (EP I, $n = 14$); EPs of the middle part of the ant colony with the highest ant activity (EP II, $n = 13$); and EPs of the lower part of the ant colony where many dead nestmates are placed on the patches (EP III, $n = 12$). Further, we studied if the bacterial/archaeal communities in established ant colonies differ significantly between the two closely related ant species. This required one bacterial/archaeal community composition per ant colony. As there was no significant difference among the different stem parts in established ant colonies, this enabled us to use the average community composition per aforementioned repeatedly sampled established ant colony. These samples were analyzed in combination with seven *A. alfari* and ten *A. constructor* colonies, from which one single patch sample per colony was collected. This dataset then resulted in overall 12 *A. alfari* and 22 *A. constructor* established ant colonies.

**PCR amplification and sequencing.** Patch sample collection, preservation and the DNA extraction process have been described in Nepel et al.[30]. The bacterial and archaeal community composition was investigated by amplifying the 16S rRNA gene for subsequent amplicon sequencing. A two-step multiplexing approach was used to amplify and barcode samples[52]. DNA-template of 10 ng or at most 4 μL was added per triplicate in the first-step PCR. The 16S rRNA gene was amplified by the primers 515F-mod (5'GTGYCAGCMGCCGCGGTAA'3)[53] and 806R-mod (5'GGACTACNVGGGTWTCTAAT'3)[54,55] including a 16 bp head sequence for subsequent barcoding[52]. A no-template control using water instead of template DNA was added in every PCR run as a contamination control. Each PCR reaction was 20 μl in volume and contained 1 × DreamTaq Green Buffer including 2 mM MgCl2, 0.2 mM dNTP mixture, 0.5 U DreamTaq DNA polymerase, 0.08 μg μL$^{-1}$ bovine serum albumin (all from Thermo Scientific, Waltham, MA, United States), 0.25 μM of each forward and reverse primers and DNA template. The following amplification program was used: 94 °C for 3.5 min followed by 22 (or 25 if DNA template <10 ng) cycles of 94 °C for 30 s, 52 °C for

45 s, 72 °C for 45 s; and final 72 °C for 10 min. Sample preparation for amplicon sequencing was done according to Nepel et al.[30]. Illumina Truseq library preparation and MiSeq sequencing was performed by Microsynth (Balgach, Switzerland) in the $2 \times 300$ cycle configuration using the MiSeq Reagent kit V3 (Illumina, San Diego, CA, United States). The raw reads were deposited into the NCBI Short Read Archive under the BioProject accession number PRJNA777006[56]. Raw MiSeq amplicon reads were processed according to the pipeline described in Herbold et al.[52], except for a revised downstream clustering of amplicon sequences. Amplicon sequence variants (ASVs) were inferred using the DADA2 package (version 1.2.0;[57]) in R (version 4.3.0;[58]). ASVs were classified against the SILVA database v138[59]. ASVs assigned to chloroplast and mitochondria were removed, with 8858 ASVs remaining. Investigated 116 patch samples contained at least 1100 and on average around 10,000 16S rRNA reads per sample.

**Statistics and reproducibility.** All subsequent data management and analyses were performed in R (version 4.3.0;[58]). Data handled and filtered based on research questions using the package phyloseq (version 1.44.0;[60]). Analysis functions originated mainly from the package vegan (version 2.6-4;[61]) unless otherwise mentioned. Graphs were plotted by using the package ggplot2 (version 3.4.2;[62]).

Absolute sequence reads were converted to relative abundances per sample for microbial community analyses. Community dissimilarity distance matrices were calculated after Bray–Curtis and displayed as principal coordinate analysis by the function capscale(). Permutational multivariate ANOVA (PERMANOVA; function adonis(); $10^4$ permutations;[63]) was used for testing significant correlations between categorical variables (patch age, ant species, developmental stage) and the variance in the community composition. Significant variables were subsequently included and visualized as constrained distance-based redundancy analysis (dbRDA) by the function capscale(). Shannon indices were determined as alpha diversity measurements by the function phyloseq::estimate_richness() and tested for significant correlations with categorical variables by Wilcoxon (function wilcox.test()) and Kruskal-Wallis (function kruskal.test()) rank sum tests.

As PERMANOVA did not show significant correlations between the bacterial community composition and the assumed patch age in multiple-sampled established ant colonies, for further analyses, one average community composition per ant colony was calculated and merged with the dataset of single-sampled established ant colonies.

For differential abundance analyses testing which taxonomic orders significantly differ between two categories (IP and YP; YP and EP; EPs of *A. alfari* and *A. constructor*) ASVs were merged at taxonomic order level (phyloseq::tax_glom()) and subsetted beforehand. Orders, which were present in less than 50% of samples, or which accounted for less than 0.05% of total reads, were dropped before differential abundance analyses. Samples with absolute read counts were centered log-ratio transformed after Monte Carlo (aldex.clr(), package ALDEx2; version 1.32.0;[64]) and Wilcoxon Rank Sum tests (ALDEx2::aldex.ttest()) were used to test every taxa for a significant shift in abundance. P-values were corrected after Benjamini-Hochberg to account for multiple testing. Only Taxa showing a mean relative read abundance of more than 0.5% across all samples were highlighted in results.

Using the function core() (package microbiome; version 1.22.0;[65]), prevalent ASVs were extracted from different datasets. Prevalent ASVs were defined as accounting for more than 0.4%

of reads in more than 50% of respective samples and were identified by the function microbiome::core_members(). For comparisons between sample types (EP: *A. alfari – A. constructor*; IP – YP – EP), the lists of prevalent ASVs were then searched for overlaps and displayed as tables. Due to the significant differences in the community composition of EPs by the ant species, we determined prevalent ASVs of *A. alfari* and *A. constructor* EPs separately and merged them finally containing all prevalent ASVs in EPs. This list of ASVs in combination with the ones from IPs and YPs were exported as a table and used for the ternary analysis showing the abundance of every prevalent ASV in the three patch types representing developmental stages of ant colonies. For the ternary analysis, the samples of the IP-YP-EP dataset were first merged by sample type (IP, YP, EP_alf, EP_con; phyloseq::merge_sample()) and read abundances were normalized by transforming them to relative abundances (phyloseq::transform_sample_counts()). Then, both EP categories were merged and a final repeated transformation to relative abundance was performed to weigh both ant species in established colonies equally. We depicted the mean community compositions as three artificial samples (IP, YP, EP). Subsequently, the dataset was filtered to contain only the list of prevalent ASVs from the previous analysis and plotted as a ternary graph (ggtern(); package ggtern; version 3.4.2;[66]). Analysis scripts are available from https://github.com/mnepel/bacteria_in_arboreal_ant_nests[67].

**Database check of identified prevalent ASVs**. To evaluate potential functional capacities of prevalent ASVs and to infer why they may be abundant in certain ant colony developmental stages, 16S rRNA sequences were checked against GenBank (NCBI) database using BLAST[68]. Subsequently, literature for the cultivated strains with the highest sequence similarity to prevalent ASVs was searched and interpreted.

**Evaluating the carbon and nitrogen content of plant tissue used for forming patches**. *Cecropia* plant parenchyma of uncolonized internodes ($n = 5$) were sampled and dried at 60 °C. The carbon and nitrogen contents of plant parenchyma were determined using an elemental analyzer (EA 110; CE Instruments, Milan, Italy) in Vienna, Austria.

**Reporting summary**. Further information on research design is available in the Nature Portfolio Reporting Summary linked to this article.

## Data availability

The raw reads and metadata were deposited into the NCBI Short Read Archive under the BioProject accession number PRJNA777006[56]. Analysis scripts and input data are available from https://github.com/mnepel/bacteria_in_arboreal_ant_nests[67]. At this repository, the source data behind the analyses and figures can be retrieved as a text file containing an ASV count table including meta data and the representative sequences, and as a spreadsheet containing Shannon indices.

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

## Acknowledgements

This work was funded by a project grant (P-31990-B to VEM) from the Austrian Science Fund (FWF). MN was supported by a DOC fellowship (24388) from the Austrian Academy of Sciences (ÖAW) and a fellowship by the Society for the promotion of the La Gamba Field Station. We thank the staff at the Estación Tropical La Gamba for great working conditions and we thank Costa Rican authorities for their permissions to conduct this research (R-046-2015-OT-CONAGEBIO, SINAC SE-GUS-PI-113-2015, INV-ACOSA-001-2016, INV ACOSA-013-2018, DGVS-481-2015, DGVS-061-2016, DGVS-088-2018). We also thank Adrián Pinto-Tomás for the warm welcome and interesting discussions every time we visited San José, and for facilitating to hold a joint Fluorescence in situ Hybridization (FISH) workshop at the Universidad de Costa Rica.

## Author contributions

D.W., V.E.M. and M.N. designed this study. V.E.M., M.N. and D.W. conducted fieldwork and collected samples. M.N. performed lab work, analyzed the data and wrote the original draft of the manuscript. All authors, including V.B.S., contributed substantially to revisions. All authors read and approved the final manuscript.

## Competing interests
The authors declare no competing interests.
