## [Peer Review File · Communications Biology]

Reviewers' comments:

Reviewer #1 (Remarks to the Author):

Review comments on manuscript 17190

This manuscript is well written and the discussion makes a lot of sense. There is a need for more details in the method section but the rest is well documented, the figures are easy to understand and pretty. The content also is novel and would belong in this journal.

Minor edits:

I.52 : Are the ants or the bacteria impacting free-living bacterial communities? Also it sounds like you're talking about the nesting space of these free-living bacteria. Try to rephrase this sentence

I.54: Specify here that you are talking about bacteria in the soil of ants' nests. Otherwise the reader wonders why you again talking about the function of bacteria in ants' nest (since you mentioned it in the paragraph before)

I.61: in N2 fixation

I.65: How can it influence the soil if it's a free-hanging nest?

I.68-69: You say that the bacterial communities are different from the free-hanging nests described above but yet you mention the three same groups for both. If you want to show the difference, maybe go to a lower taxonomical level. Same thing when talking about the nests inside the tree

I.73: What is the difference in structure

I.82: The reference for bacteria should be superscript

I.358: In line 348, you mention that 64 plants were investigated so why only 31 plants were sampled?

I.361: Why are there such large differences between the number of samples of IP, YP, and EP?

I.363: Why did you keep this species is it only had IP?

I.365: 17 colonies for each species? If not, how many colonies were sampled in each species? And how many of each type of EP was sampled for each species

I.371-376: This is confusing. It's hard to understand what the purpose of this average community composition is (or why it's in the sampling part of the methods) and of the additional sampling

I.381: Please add the primers sequences

I.394: Why OTUs and not ASVs?

I.396-400: Some of those sentences belong in the results section not in the methods, especially since the results section starts with the exact same sentences

Reviewer #2 (Remarks to the Author):

Marilia Bitar

This is a very interesting study where the authors investigated the patches bacterial communities along the ant colony life cycle and between closely related ants, in an obligatory ant-plant mutualism. They found that the maintenance of these bacterial communities is driven by ecological processes (ecological succession and substrate-caused bottleneck) related to the colony developmental stages and to the species. The authors used consistent analyses to prove these processes and are bringing novelty to questions about the interactions between ant-plants-bacteria. Follow some comments:

Lines 43 – 45: Some works investigate this. I recommend putting some references.

Lines 85 – 86: I would shed light on the question: Do these patches have any benefit to the mutualistic host? What type?

Lines 108 – 109: What about fungi and nematodes? Perhaps the term "bacterial patch community" is

confusing.

Line 110: What do you mean by "heterogeneous" bacterial community composition? Would it be the diversity between the samples? Heterogeneity concerns the whole environment (in this case, the spots?). Why is it heterogeneous if you find low diversity in early patches?

Lines 123 – 125: So, each colony has its own microbiome? You found this result in other analyses, it might be important to say in the discussion/conclusion that bacterial communities in patches have some specificity in each colony.

Lines 148 – 149: How was it observed? I don't see why put this result if it's not statistically significant.

Lines 158 – 159: This part is confusing. Did you analyze 34 or 17 established ant colonies?

Lines 217 – 218: Perhaps the diversity of the bacterial community could be related to the defense strategies of the larvae and brood. In addition, a greater abundance of Actinobacteria was found in earlier stages.

Lines 237 – 240: How did you observe this evidence?

Lines 245 – 246: Do Azteca ants have metapleural glands?? Some arboreal ants don't have. See: <https://doi.org/10.1111/j.1469-185X.2010.00170.x>
Hölldobler, B., & Engel-Siegel, H. (1984). On the metapleural gland of ants. *Psyche*, 91(3-4), 201-224.

Lines 255 – 257: Could IP have more bacteria from the plant since they don't have so many bacteria from outside transported by workers?

Lines 301 – 302: How can EP be highly diverse and homogeneous?

Lines 302 – 304: Have you investigated whether any of these OTUs are good for the host plant?

Lines 308 – 309: When you said, "inhabit the same host plants", are you referring to the same individuals?

Lines 356 – 357: Although some of the methodologies are described in Nepel et al. (2022), I strongly recommend that you include a photo of these spots in this article.

Reviewer #3 (Remarks to the Author):

This manuscript describes a survey of the microbial communities in Azteca ant genus patches that were taken at different phases of the life cycle of ants. The authors have demonstrated temporal changes in the diversity and community composition of the patches. These differences were shared by the three species that were surveyed in this study. As the authors claim in the abstract this is an interesting study that opens many questions for future studies, however, the study is limited in scope. Moreover, the analysis and presentation of the results could be improved as detailed below:

Abstract

Most of the abstract introduces the study, but there is very little information about the results. The authors present the hypothesis that drove the study but not the results. The authors only claim that the study provides key information for future studies, yet they do not reveal what that key information might be. The abstract should be rewritten with more emphasis on the results and less on introducing

the study.

Introduction

The second and third paragraphs should be substituted. The authors should first describe the studied ant genus and then describe the associated bacteria. The description of the bacteria in the second paragraph seems out of context and would make better sense after the description of the biology of the ant genus in the third paragraph.

The knowledge gap that advised this study is not clear. What was the motivation for this study. The authors pose the question of the temporal dynamics of the bacteria/archaea community during the life cycle of a colony. However, they do not describe the background. In other words, the authors do not describe other studies that reported temporal changes in the microbial community of ants.

The hypotheses presented are not clear: (I) Do the authors mean that the same species are found in all stages of development of the ant genus or that each stage is typified by a different community?; (II) Do the authors mean that there are no temporal changes in the composition of the patch community? If so, then for how long? Why would they expect temporal changes? Does their diet or environmental conditions change?; (III) The syntax of the third hypothesis is confusing and should be edited. If I understand this, then this hypothesis negates the previous hypotheses. What do the authors mean by 'some variation' do they mean significant differences or insignificant differences? The hypothesis is not clear.

Results

The authors have sampled three different species of the *Azteca* genus. Why did they assume that the community composition in the patches of the different species would be similar? Did they hypothesize that the patch communities are equal or do they want to test whether they differ? In the introduction, they do not describe the three different species that were tested here, and it is not clear what might be the differences or similarities between the selected species. The sampling design is not clear and does not fit well with the proposed hypotheses.

The authors claim that 99.9% of the total reads were bacteria; however, it was shown that 515F and 806R primers are biased against some Archaeal phyla specifically, Crenarchaeota and Thaumarchaeota. Therefore, the results obtained should be reported with caution.

Figure 1a. The composition of the bacterial community of *A. alfari* seems random, some samples are dominated (~80% it seems) by one family while others show higher variability. Are the authors sure that the single-family samples are not a result of contamination? It is unexpected that the community is heavily dominated by a single family. Furthermore, the authors do not acknowledge this variability in the replicates while it could explain the clustering in Figure 1b.

It is difficult to follow this bar plot and understand the composition of each community of species. I strongly suggest presenting the average relative abundance of each community of ant species and present the detailed bar graph in the supplementary information. This could be applied to all the bar plots presented in the manuscript. It is hard enough to follow bar plots in general, but having to follow such differences in the communities is very confusing.

There seem to be repetitions in the presented data (alpha diversity in Figures 3, 4 and 5). The authors seem to present the same results in different analyses. To serve the narrative of the study, I would think that Figures 5 and 6 will suffice. Figure 5 shows the temporal changes in the microbial communities, at the genus and species level. The authors may consider separating the results to two figures, one showing the changes at the genus level and the other at the species level (including averages of the community composition).

Showing the community composition at both the class and order levels is redundant and confusing. The authors should choose one taxonomic level and demonstrate the changes using that taxonomic level, while in the supplementary they can show other taxonomic levels and mention that briefly in the text.

Discussion

The discussion is lengthy and repetitive. The authors should be more concise in their description of the results.

Materials and Methods

The authors should add a figure (could be in supplementary) that describes the life cycle of the ant using photos or drawings. The provided description does not depict the life cycle of species well and it would be beneficial to visualize it to the reader.

For the read analysis, the authors did not use the prevailing software dada2 that was shown in multiple studies to be superior to the older method that was used in this study. Moreover, the authors chose to use the older version of QIIME. It is not clear why the authors decided to use these older and less reliable methods that were proven to reflect microbial communities in different environments with less accuracy.

It would have been interesting if the authors had added the measure of bacterial abundance to their analysis. It might be tricky to normalize the bacterial abundance, but it could be related maybe to surface measures or sample volume? Furthermore, measures of patch size may also have been relevant in this study.

Reviewer #1 (Remarks to the Author):

This manuscript is well written and the discussion makes a lot of sense. There is a need for more details in the method section but the rest is well documented, the figures are easy to understand and pretty. The content also is novel and would belong in this journal.

Minor edits:

1	I.52 : Are the ants or the bacteria impacting free-living bacterial communities? Also it sounds like you're talking about the nesting space of these free-living bacteria. Try to rephrase this sentence
	>It is still unknown to which extend ants or their "ecto- and endosymbionts" are impacting free-living bacterial communities in ants' nesting space. However, in the mentioned sentence, we aim to highlight that besides ant-body associated bacteria also free-living bacteria in ants' nesting space may play an essential role for the ant colonies. L56-58: We rephrased the sentence to clarify this.
2	I.54: Specify here that you are talking about bacteria in the soil of ants' nests. Otherwise the reader wonders why you again talking about the function of bacteria in ants' nest (since you mentioned it in the paragraph before)
	>We agree that this sentence does not adequately introduce recent studies. L56-68: We rewrote this paragraph and included more details of recent studies (ant-plant association and sample type)
3	I.61: in N2 fixation
	>L68: we implemented this suggestion.
4	I.65: How can it influence the soil if it's a free-hanging nest?
	>Apparently, Azteca trigona ants influence the bacterial community composition in soil underneath their nest by dropping organic material. However, this was not the intended message of the sentence. Instead, we wanted to stress that A. trigona ants influence their bacterial and fungal nest microbiome. However, after restructuring the introduction (suggested by Reviewer #3), we decided to remove this citation. Since the paragraph now focuses on ants living inside host trees, the study investigating free-hanging A. trigona nests does not apply anymore.
5	I.68-69: You say that the bacterial communities are different from the free-hanging nests described above but yet you mention the three same groups for both. If you want to show the difference, maybe go to a lower taxonomical level. Same thing when talking about the nests inside the tree
	>We thank the reviewer for pointing this out and agree that mentioning classes here does not show any significant community differences compared to other studies. However, as mentioned in the reply to the former comment, we decided to remove this citation, as it does not fit in this paragraph anymore after restructuring the introduction.
6	I.73: What is the difference in structure
	> As difference in structure, we intended to describe the community differences of N ₂ -fixing bacteria in ant nests. However, due to revising this paragraph and focusing it on the general microbiome, we removed this sentence.
7	I.82: The reference for bacteria should be superscript
	>L96: We now changed the citation to be superscript.

8	I.358: In line 348, you mention that 64 plants were investigated so why only 31 plants were sampled?
	>We agree that this was not properly outlined in the methods section. In total we sampled 65 Cecropia plants, not 64 (we apologize for this typo). Of these 65 plants, 31 were young Cecropia saplings containing recently founded and young ant colonies. In addition to these 31 plants, we sampled 34 Cecropia trees that were inhabited by established ant colonies. We now clarified in L346 that we sampled 65 plants of different sizes – 31 saplings and 34 trees. We now provide additional information on these 34 trees and their established ant colonies in L364-378.
9	I.361: Why are there such large differences between the number of samples of IP, YP, and EP?
	>Differences in the sample sizes were due to the biology of this ant-plant association. To study this ant-plant association in its natural habitat in our study area, one depends on field work in the tropics, which is not always predictable. Also, sampling the younger stages (IPs and YPs) is particularly challenging, as described below. There are two interconnected reasons for the variable sample size of IP and YP. First, the approach for our study was destructive by cutting plants. Therefore, one cannot sample unlimited Cecropia plants, if one wants to maintain the population in the study area. Second, only after cutting and opening Cecropia saplings one can see if or how many internodes are inhabited, in which colony foundation stage the ants are, and also by which Azteca species they are inhabited. As it is unpredictable which ant species and at what stage will be found in the cut plant, it is hardly possible to achieve similar sample sizes per ant colony developmental stage and ant species. After cutting more than 100 Cecropia saplings in total, only a fraction of internodes was colonized by ants and could be used for our study. Furthermore, of the colonized internodes many queens were found dead, which is not surprising for this ant-plant association but limiting the sample size. Ant queens were also found with hardly any patch material, as they just recently entered the internode (also noticeable by the appearance of the plant tissue inside the hollow internode). Thus, to reach the high number of samples in our study, we had to conduct multiple field trips and let the plant as well as ant populations recover in between. L349: We added a sentence addressing the variation in sample size.
10	I.363: Why did you keep this species is it only had IP?
	>Although only the initial colonization stage of A. xanthochroa was found in our study area, we aimed to present as much information about the system as possible. We decided not to discard these samples, as a third ant species strengthens the statistical analyses and emphasize our finding that the bacterial community composition in IPs is low diverse and not correlating with the ant species.
11	I.365: 17 colonies for each species? If not, how many colonies were sampled in each species? And how many of each type of EP was sampled for each species
	>We rephrased this paragraph (L364-378) and added more details to clarify the sample sizes per ant colony developmental stages and ant species. L364, we sampled 34 Cecropia trees which were inhabited by established A. alfari (n=12) and A. constructor (n=22) ant colonies.
12	I.371-376: This is confusing. It's hard to understand what the purpose of this average community composition is (or why it's in the sampling part of the methods) and of the additional sampling
	>We revised this paragraph (L364-378) to improve clarity of our sampling design. Studying established ant colonies, we addressed two research questions. (I) Is the composition of the bacterial/archaeal community consistent within the patches of the same established Azteca colony? We collected microbial patches from A. alfari (n=5) and A. constructor (n=12) colonies

	separately from up to three stem parts representing different ant colony activity and patch age. (II) Is the bacterial/archaeal community composition in patches of two closely related Azteca colonies species-specific? For that question, we required to have only one bacterial/archaeal community composition per established ant colony. We therefore calculated an average community composition per aforementioned repeatedly sampled established ant colony. Additionally, seven A. alfari and ten A. constructor colonies were sampled once to increase the sample size for answering this question.
13	I.381: Please add the primers sequences
	>L385: We now added the primer sequences.
14	I.394: Why OTUs and not ASVs?
	>We initially decided to take the 97% OTU clustering and not the ASV approach, as we aimed to keep this study as comparable as possible to our earlier study (Nepel et al. 2022), where we investigated the diazotrophic community composition in the same biological samples using a 97% OTU clustering. However, as this was a recurring comment from the reviewers, we have now processed our sequencing data using DADA2 to infer ASVs. We reran our entire community analysis and subsequently updated the manuscript main text including figures and tables. Using the ASV approach instead of the 97% OTU clustering, the outcome of our analysis has not changed.
15	I.396-400: Some of those sentences belong in the results section not in the methods, especially since the results section starts with the exact same sentences
	>We agree that some sentences are redundant and belong in the results section. We revised this paragraph part by removing and rephrasing sentences (L400-402).

Reviewer #2 (Remarks to the Author):

Marilia Bitar

This is a very interesting study where the authors investigated the patches bacterial communities along the ant colony life cycle and between closely related ants, in an obligatory ant-plant mutualism. They found that the maintenance of these bacterial communities is driven by ecological processes (ecological succession and substrate-caused bottleneck) related to the colony developmental stages and to the species. The authors used consistent analyses to prove these processes and are bringing novelty to questions about the interactions between ant-plants-bacteria.

Follow some comments:

1	Lines 43 – 45: Some works investigate this. I recommend putting some references.
	>After the mentioned sentences, we cited at least eight of these indicated studies including more details. However, it seems that the link to these references was not apparent to the reader. Therefore, we now cite these studies already earlier (L44-45) and subsequently mention them in more detail later in this paragraph.
2	Lines 85 – 86: I would shed light on the question: Do these patches have any benefit to the mutualistic host? What type?
	>We agree that the role of patches for the host plant is an interesting, yet not fully answered question in the Azteca-Cecropia association. For instance, studies investigated nutrient flow from the Azteca colony to the Cecropia host plant (Sagers et al. 2000, DOI:10.1007/PL00008863; Dejean et al. 2012, DOI: 10.1016/j.crv.2012.01.002), which may be facilitated via patches. Yet, specialized plant tissue for taking up nutrients underneath patches was not visible microscopically, nor fungal hyphae that grow from patches into plant tissue. However, our study focuses on the relation between the ant colony and the bacterial community in patches. As such, we consider the potential benefit to the host plant as additional and interesting, but not crucial information, which would make the paragraph more complex. Therefore, we decided not to add this information in the " Azteca-Cecropia paragraph", but in an earlier paragraph. In L54-55, we added that ants often provide nutrients to the host plant in form of their debris.
3	Lines 108 – 109: What about fungi and nematodes? Perhaps the term "bacterial patch community" is confusing.
	>In this study, we focused on the bacterial/archaeal community and did not investigate fungi or nematodes. We agree that these lines can be differently understood and rephrased L118-120 for better clarification. Because of the vast dominance of bacterial reads, all community patterns are solely related to bacteria and therefore we refer to the archaeal/bacterial community as bacterial community. This has now been explained in the manuscript.
4	Line 110: What do you mean by "heterogeneous" bacterial community composition? Would it be the diversity between the samples? Heterogeneity concerns the whole environment (in this case, the spots?). Why is it heterogeneous if you find low diversity in early patches?
	>This sub header relates to the strong heterogeneity in the bacterial community composition across IP samples (L131-143), even within the samples of one ant species. Heterogeneous community composition describes dissimilar community compositions across samples with taxa (or in our case taxonomic orders) strongly varying in their abundance. These taxa/orders are dominating in certain samples and are hardly present in others. Low alpha diversity is not necessarily related to high heterogeneity, though, high heterogeneity may be easier visible in plots if the samples are low diverse. We have now specified that heterogeneity refers to the

	bacterial community composition (L133) and also modified the header for clarity.
5	Lines 123 – 125: So, each colony has its own microbiome? You found this result in other analyses, it might be important to say in the discussion/conclusion that bacterial communities in patches have some specificity in each colony.
	>Although this may be true to a certain extent, we find it difficult to make this statement. We can show, for early colony development stages, that there is no significant correlation between the variation of the community composition and the ant species. Nevertheless, a few microbiomes show a high similarity, even from different ant species (indicated in Figure 2b). Furthermore, postulating that each colony has its own microbiome could be also interpreted as each ant colony having its own stable microbiome throughout the ant colony life cycle, which we did not find evidence for. Bacterial community composition and beta diversity pattern change from early to late colony developmental stages. Thus, we refrained from adding this statement to our discussion/conclusion.
6	Lines 148 – 149: How was it observed? I don't see why put this result if it's not statistically significant.
	>We agree that this is misleading and revised the sentence for clarity (L163-164). In our opinion it is still worth mentioning that there is no significant difference in alpha diversity between patches of different age within an established ant colony. Based on the process of patch formation in established ant colonies and potential substrate differences, an increase in alpha diversity from young to old patches could have been possible.
7	Lines 158 – 159: This part is confusing. Did you analyze 34 or 17 established ant colonies?
	>In this revised sentence (L174-175) and the rephrased method section (L364-378) we now better clarify the sampling effort for established ant colonies. In total, patches of 34 established ant colonies were sampled. In 17 of these ant colonies, patches were collected separately from different locations within the ant nesting space, referring to different patch ages (EPI, EP II and EP III). These samples were used to investigate if the bacterial community composition varies in patches of different ages in the same established ant colony. From the other 17 established ant colonies only one patch sample was collected. The bacterial community compositions of all 34 established ant colonies were used to investigate if the bacterial communities differ significantly between the two closely related ant species.
8	Lines 217 – 218: Perhaps the diversity of the bacterial community could be related to the defense strategies of the larvae and brood. In addition, a greater abundance of Actinobacteria was found in earlier stages.
	>We agree that one potential function of patches might be to cultivate certain bacteria producing anti-microbial metabolites, which can be used to protect larvae/brood against pathogens. However, our data does not indicate that this shapes the bacterial community composition. First, both early ant colony developmental stages (IPs and YPs) are limited to a single internode containing the ant queen, larvae/brood, patch and only in young ant colonies also first worker ants. We would argue that protecting the brood would be the main function in both stages, however we see an increase of alpha diversity and changes in the bacterial community composition. Second, the relative abundance of Actinobacteria needs to be seen with caution. Whereas the average relative abundance of Actinobacteria is higher in IPs than in YPs, Figure 2a (former Fig. 1a) shows that this is because Actinobacteria dominate in six out of 42 IP samples. Also, a decrease in relative abundance does not necessarily mean a decrease in absolute abundance. As the alpha diversity increases from IPs via YPs to EPs, the abundance of other taxonomic groups appears to increase, instead of a decrease in Actinobacteria. Quantitative PCR would be needed to confirm differences in absolute abundance. Third, the order Corynebacteriales being abundant in IPs and YPs is not the commonly known Actinobacterial order to produce anti-microbial metabolites. However, this does not necessarily

	exclude their functionality of protecting the brood. Thus, we decided not to add the suggested aspect into our discussion.
9	Lines 237 – 240: How did you observe this evidence?
	>Most of this information is derived from observations of ant behavior and visual appearance of inhabited internodes. Mass spectrometry was used to determine the carbon/nitrogen ratio of Cecropia parenchyma commonly used for forming patches. We have now added the average C/N ratio for plant parenchyma in our manuscript (L254). Additionally, we included method details and mass spectrometry measurements in the supplementary information file (supplementary Table 2).
10	Lines 245 – 246: Do Azteca ants have metapleural glands?? Some arboreal ants don't have. See: https://doi.org/10.1111/j.1469-185X.2010.00170.x Hölldobler, B., & Engel-Siegel, H. (1984). On the metapleural gland of ants. Psyche , 91(3-4), 201-224.
	>Although it is correct that some arboreal ant species evolutionary lost their metapleural glands, there are other arboreal ants that still have them. As metapleural glands were present in arboreal Azteca velox (Walker & Hughes 2011, DOI:10.1111/j.1365-2311.2011.01312.x) we do not expect the absence in our Azteca species.
11	Lines 255 – 257: Could IP have more bacteria from the plant since they don't have so many bacteria from outside transported by workers?
	>Based on our data, we find it unlikely that the detected bacteria composition in IPs originated mainly from plant tissue inside the plant internode. We rather hypothesize that the queen entering the plant is the main vector introducing bacteria in the early ant colony developmental stage. We did not see any indication that the plant parenchyma inside the internode contained many bacteria potentially shaping or dominating the community composition. We were not able to amplify the 16S rRNA gene in DNA extracts from dried parenchyma of uncolonized internodes; however, we do not have enough evidence to claim that this parenchyma tissue is sterile. If bacteria were present in the parenchyma within the plant, these bacteria would be present throughout this tissue. Subsequently, as this scratched-off parenchyma from the inside wall of the plant internode is the only substrate used to form patches in early ant colonies, we would have to detect these bacteria constantly in the community composition of IPs and YPs. As the bacterial community compositions of these patches are very heterogeneous across samples with differing dominating taxa, these potential parenchyma-associated bacteria – if existing – seem to show only low relative abundance.
12	Lines 301 – 302: How can EP be highly diverse and homogeneous?
	>In our manuscript we used “low/high diverse” to characterize the community composition of single samples (equivalent to alpha diversity). The wording “heterogeneous” and “homogeneous” referred to the community variability across colonies in the same development stage (=patches of the same type). Therefore, IPs are low diverse and heterogeneous, as mostly one to two taxonomic orders are dominating the community composition of an IP and at the same time the variability across samples is high with different orders dominating in different IPs. In contrast, EPs are highly diverse and rather homogeneous, as every EP contains many taxonomic orders and at the same time the variability across samples is rather low with similar relative abundances of these orders. We hope to increase clarity by writing “across” in L236 and in figure captions.
13	Lines 302 – 304: Have you investigated whether any of these OTUs are good for the host plant?
	>We agree that it would be very interesting to know the function of these OTUs, or at least classifying them as beneficial for the ants and/or the host plant. However, we concluded that it is not feasible in this study, as it is very difficult to assign a potential function based on a 16S

	rRNA gene sequence. In addition, it is challenging to further assess if this potential function would be beneficial for the host plant. Thus, we decided to only focus on the bacterial- and ant-side of this association.
14	Lines 308 – 309: When you said, "inhabit the same host plants", are you referring to the same individuals?
	>In this case, the sentence in our manuscript aimed to emphasize that the investigated ant species are closely related and inhabit the same habitat and host plant species. Similar to other parts of this discussion, this paragraph has been rephrased now. However, for your information, the ant queens literally compete for the same host plants and often inhabit the same plant individuals. This can be observed in the early ant colony developmental stages. As soon as one colony has enough workers to enlarge their nesting space by connecting additional internodes, the other ant queens get killed.
15	Lines 356 – 357: Although some of the methodologies are described in Nepel et al. (2022), I strongly recommend that you include a photo of these spots in this article.
	>We thank the reviewer for this comment and agree that it is beneficial to add an illustration for better visualization. We decided to add a schematic drawing of the system as Figure 1 to better display the differences between the three ant colony developmental stages.   Figure 1: Schematics of the three investigated ant colony developmental stages and their corresponding microbial patches.

Reviewer #3 (Remarks to the Author):

This manuscript describes a survey of the microbial communities in Azteca ant genus patches that were taken at different phases of the life cycle of ants. The authors have demonstrated temporal changes in the diversity and community composition of the patches. These differences were shared by the three species that were surveyed in this study. As the authors claim in the abstract this is an interesting study that opens many questions for future studies, however, the study is limited in scope. Moreover, the analysis and presentation of the results could be improved as detailed below:

1	Abstract Most of the abstract introduces the study, but there is very little information about the results. The authors present the hypothesis that drove the study but not the results. The authors only claim that the study provides key information for future studies, yet they do not reveal what that key information might be. The abstract should be rewritten with more emphasis on the results and less on introducing the study.
	>We agree that the initial abstract focused more on background information than on results. We have revised the entire abstract accordingly by focusing on our findings.
2	Introduction The second and third paragraphs should be substituted. The authors should first describe the studied ant genus and then describe the associated bacteria. The description of the bacteria in the second paragraph seems out of context and would make better sense after the description of the biology of the ant genus in the third paragraph.
	>Considering this comment, we rephrased several paragraphs of the introduction to improve the reading flow. Our revised line of thought improves the introduction of our manuscript: after (I) mentioning the importance of ant body-associated bacteria enabling ants to colonize the tropical canopy, we continue with (II) introducing studies investigating free-living bacteria in arboreal ant nests. (III) This microbiome has been shown to be influenced by ants. Yet, (IV) there are many knowledge gaps of bacterial communities inside plants inhabited by mutualistic ants. (V) To investigate the dynamics of the bacterial community during the life cycle of arboreal ant colonies, we chose the Azteca-Cecropia association and addressed certain research questions.
3	The knowledge gap that advised this study is not clear. What was the motivation for this study. The authors pose the question of the temporal dynamics of the bacteria/archaea community during the life cycle of a colony. However, they do not describe the background. In other words, the authors do not describe other studies that reported temporal changes in the microbial community of ants.
	>We agree that we were not able to adequately point out the knowledge gap. In the newly revised introduction, we devote an entire paragraph (L79-90) to highlight unresolved knowledge gaps. To investigate the bacterial community composition in arboreal ant nests and answer certain research questions, we chose to study the Azteca-Cecropia association (L91-111). To the best of our knowledge, we could not identify any studies which reported temporal microbiome changes in nests of arboreal ants. Other studies e.g. investigating the gut microbiome of ants are in our opinion not meaningful for investigating the research questions and were therefore not mentioned.
4	The hypotheses presented are not clear: (I) Do the authors mean that the same species are found in all stages of development of the ant genus or that each stage is typified by a different community?; (II) Do the authors mean that there are no temporal changes in the composition of the patch community? If so, then for how long? Why would they expect temporal changes? Does their diet or environmental conditions change?; (III) The syntax of the third hypothesis is confusing and should be edited. If I understand this, then this hypothesis negates the previous

	hypotheses. What do the authors mean by ‘some variation’ do they mean significant differences or insignificant differences? The hypothesis is not clear.
	>We appreciate the critical review of our hypotheses and agree that they were unclear. Thus, we revised the text to facilitate a better understanding. Considering all knowledge gap- and hypotheses-related comments of reviewer #3, we decided to rephrase “testing of hypotheses” to “addressing research questions”, as we believe this increases the clarity of our research goals. L103-109 contains our revised research questions.
5	Results The authors have sampled three different species of the Azteca genus. Why did they assume that the community composition in the patches of the different species would be similar? Did they hypothesize that the patch communities are equal or do they want to test whether they differ?
	>By revising our manuscript, we switched to asking research questions instead of testing hypotheses to prevent such ambiguity. We did not hypothesize similar bacterial communities in patches of different ant species. We assumed the opposite. According to former hypothesis (I) we expected ant species-specific community composition in all ant colony developmental stages, as ants are shaping their surrounding microbiome (see introduction L69-78). In other words, if looking at early or established ant colonies, we expected to see a similar bacteria community in patches of the same ant species, but a significantly different community compared to patches of another ant species.
6	In the introduction, they do not describe the three different species that were tested here, and it is not clear what might be the differences or similarities between the selected species. The sampling design is not clear and does not fit well with the proposed hypotheses.
	>It is correct that we did not mention the three Azteca species in the introduction. We added a general sentence (L101-103) noting that we performed 16S rRNA amplicon sequencing in patches of three colony developmental stages in up to three different Azteca species. A description of the developmental stages referring to the schematic Figure 1 follows in the following paragraph (L113-116). We did not add more details about the three ant species, as there is hardly any information available. A. alfari has the smallest queen size of Azteca species and tends to have the weakest defense of the host tree. These three Azteca species appear to overlap at our study site. Whereas established A. alfari and A. constructor colonies can be found in Cecropia trees, only initial A. xanthochroa colonies could be found. This indicates that our study site is within the A. xanthochroa distribution area, but conditions are suboptimal for colony foundation. We rephrased our hypotheses to research questions for better understanding and added more details and clarified the sampling design in the methods section.
7	The authors claim that 99.9% of the total reads were bacteria; however, it was shown that 515F and 806R primers are biased against some Archaeal phyla specifically, Crenarchaeota and Thaumarchaeota. Therefore, the results obtained should be reported with caution.
	>We agree that the initial primers 515F and 806R are biased against some Archaeal phyla. However, we used the modified version of this primer pair as described in the methods. The primers 515F-mod and 806R-mod were designed to better cover biased taxa, like these Archaeal phyla (Walters et al. 2015, DOI:10.1128/mSystems.00009-15.). According to SILVA primer check, this modified primer pair covers around 85% of Bacteria and 85% of Archaea in the current SILVA database. Therefore, we see similar primer biases in both groups. However, we rephrased this sentence (L117-120) and explained that we call the microbiome “bacterial community” as all patterns and correlations are linked to the bacterial community.
8	Figure 1a. The composition of the bacterial community of A. alfari seems random, some samples are dominated (~80% it seems) by one family while others show higher variability. Are the authors sure that the single-family samples are not a result of contamination? It is unexpected

	that the community is heavily dominated by a single family. Furthermore, the authors do not acknowledge this variability in the replicates while it could explain the clustering in Figure 1b.
	>We did our best to prevent contaminations during sampling in the field and subsequent wet lab work. No-template controls in molecular processes did not indicate systematic contaminations. Sequencing these controls resulted in very few reads, which did not match taxa dominating single bacterial communities in IPs. The observation that different taxonomic orders dominate different patch samples also does not support the occurrence of contaminations. Furthermore, YPs are very comparable to IPs, but don't show orders dominating single patches. In both cases (IPs and YPs): (I) their ant colonies are limited to one internode each, (II) both were found in the same plant individual, therefore were sampled at the same time, and (III) have approximately the same size therefore one colony consisted of one patch, which was entirely used for DNA extraction und subsequent molecular work. Additionally, the DNA concentration of nucleic acid extracts tend to be higher in YPs than in IPs. Thus, in our opinion the higher variability in IPs has biological reasons. The time frame for initial ant colonies is quite broad, covering recently entered ant queens until right before workers hatch (likely several weeks). It is likely that we see different bacterial variability in IPs because they are from different succession timepoints within the early ant colony developmental stage. Very recently formed patches are expected to contain only low bacterial biomass and may therefore show mainly one dominating order which is probably able to degrade the nutritionally limited substrate. This would have to be tested in a follow-up study focusing on initial ant colonies only. Also, removing subjectively low diverse initial ant colonies, because the respective IPs are dominated by certain orders, would change Figure 2b (former Fig. 1b), but would unlikely change the statement of non-ant species-specific community composition.
9	It is difficult to follow this bar plot and understand the composition of each community of species. I strongly suggest presenting the average relative abundance of each community of ant species and present the detailed bar graph in the supplementary information. This could be applied to all the bar plots presented in the manuscript. It is hard enough to follow bar plots in general, but having to follow such differences in the communities is very confusing.
	>We thoughtfully assessed our possibility to reduce the content in our bar plots and focus on the most important message. However, we concluded that visualizing the community variability within an ant species is an important result of our study and crucial for the discussion. Depicting e.g. Figure 2a (former Fig. 1a) with only three columns, instead of showing one bar per ant colony, would imply that the communities within an ant species are similar. Subsequently, it could result in a stronger focus of the reader on bacterial community differences between ant species, which are statistically not significant due to the high variability within ant species. Thus, we decided that we want the reader to rather recognize the presence or absence of color patterns in our bar plots than the detailed community composition and kept the bar plots as they are.
10	There seem to be repetitions in the presented data (alpha diversity in Figures 3, 4 and 5). The authors seem to present the same results in different analyses. To serve the narrative of the study, I would think that Figures 5 and 6 will suffice. Figure 5 shows the temporal changes in the microbial communities, at the genus and species level. The authors may consider separating the results to two figures, one showing the changes at the genius level and the other at the species level (including averages of the community composition). Showing the community composition at both the class and order levels is redundant and confusing. The authors should choose one taxonomic level and demonstrate the changes using that taxonomic level, while in the supplementary they can show other taxonomic levels and mention that briefly in the text.
	>We agree that the figures contained some redundancies and modified Figure 6 (former Fig. 5) by moving two graphs (5b,e) into the supplementary information. Alpha diversity indices shown in former Figure 4b could also be found in former Figure 5b. As former Figure 5b (showing alpha

diversity indices per ant species) did not fit very well with the rest of the Figure (all showing data for whole *Azteca* genus), and since it is not critical for the main manuscript, we decided to move former Figure 5b into the supplementary. Also, we agree that former Figure 5d showing the community composition on class level is redundant, as the classes can be retrieved from Figure 6c (former Fig. 5e). We moved former Figure 5e to the supplementary as well. The mentioned alpha diversity dataset used for Figure 4b (former Fig. 3b) focuses on a specific study aspect and is only a subset of the datasets used for Figure 5 and Figure 6 (former Fig. 4 und 5). Therefore, we kept this data presentation in the current version of the manuscript.

Updated Figure 5: Changing bacterial community composition, alpha and beta diversity along developmental stages of ant colonies, including patches of initial (IP), young (YP) and established (EP) ant colonies.

11 Discussion
The discussion is lengthy and repetitive. The authors should be more concise in their description of the results.

>We revised the discussion and removed repetitive parts. We further streamlined paragraphs to reduce the overall length. We hope that this improves the reading flow of our manuscript.

12 Materials and Methods
The authors should add a figure (could be in supplementary) that describes the life cycle of the ant using photos or drawings. The provided description does not depict the life cycle of species well and it would be beneficial to visualize it to the reader.

>We agree that it is beneficial to add illustrations for better visualization and thus added a schematic drawing as Figure 1 in our manuscript. It better displays the differences between the three ant colony developmental stages.

Figure 1: Schematics of the three investigated ant colony developmental stages and their corresponding microbial patches.

13	For the read analysis, the authors did not use the prevailing software dada2 that was shown in multiple studies to be superior to the older method that was used in this study. Moreover, the authors chose to use the older version of QIIME. It is not clear why the authors decided to use these older and less reliable methods that were proven to reflect microbial communities in different environments with less accuracy.
	>We agree that studies have been shown DADA2 to be superior to the older method. Yet in our opinion, it is not a settled conclusion that ASVs are inherently “better” than the 97% clustering approach (see Schloss 2021, DOI:10.1128/msphere.00191-21). However, we decided to reprocess our sequencing data using DADA2 to infer ASVs. Using ASVs also serves better reproducibility if our data was to be analyzed by other scientists, which we would like to support as well. We further reran our entire community analyses (using updated versions of R software and packages) and subsequently modified the manuscript main text including figures and tables. The outcome of our analysis remained unchanged when using the ASV approach instead of the 97% OTU clustering. As QIIME was only used for sorting MID as part of an established read processing pipeline and irrelevant in the read processing of our study (Herbold et al. 2015), it is inconsequential whether QIIME 1 or QIIME2 is used. Therefore, we refrained from rerunning the read processing using QIIME2.
14	It would have been interesting if the authors had added the measure of bacterial abundance to their analysis. It might be tricky to normalize the bacterial abundance, but it could be related maybe to surface measures or sample volume? Furthermore, measures of patch size may also have been relevant in this study.
	>We agree that using quantitative measures would have been a nice addition to this study. Unfortunately, we were limited in biological material especially from early ant colony developmental stages to perform additional molecular assays for this particular study. For instance, we would have liked to obtain qPCR results to complement the relative read abundances data. However, we were not able to obtain metadata (such as volume/size of patches) that would have been needed to normalize such quantitative data. Our main focus during field work was to preserve patch material as quickly as possible in RNAlater, as other related projects (such as Nepel et al. 2022) required these samples to be amenable for RNA (transcript level) work. Therefore, we documented only a limited number of metadata, but plan to extend on this aspect in future studies.

REVIEWERS' COMMENTS:

Reviewer #1 (Remarks to the Author):

The manuscript after revision is in a very good place and I am satisfied with the answers from the authors. I especially really appreciate the Figure 1 schematics, it really helps the reader understand your study system.

Reviewer #2 (Remarks to the Author):

This is a very interesting study where the authors investigated the patches bacterial communities along the ant colony life cycle and between closely related ants, in an obligatory ant-plant mutualism. They found that the maintenance of these bacterial communities is driven by ecological processes (ecological succession and substrate-caused bottleneck) related to the colony developmental stages and to the species. The authors used consistent analyses to prove these processes and are bringing novelty to questions about the interactions between ant-plants-bacteria.

I believe the authors of the manuscript have done a good review of the material and have appropriately addressed my questions. I also believe that this study will be a valuable publication in this journal.